# Different Ultrasound Exposure Times Influence the Physicochemical and Microbial Quality Properties in Probiotic Goat Milk Yogurt

**DOI:** 10.3390/molecules25204638

**Published:** 2020-10-12

**Authors:** Karina Delgado, Carla Vieira, Ilyes Dammak, Beatriz Frasão, Ana Brígida, Marion Costa, Carlos Conte-Junior

**Affiliations:** 1Department of Food Technology, Faculdade de Veterinária, Universidade Federal Fluminense, Niterói 24230-340, Brazil; karinafrensel@gmail.com (K.D.); carlavieira2002@yahoo.com.br (C.V.); beatrizfrasao90@gmail.com (B.F.); marioncosta@ufba.br (M.C.); 2Food Science Program, Instituto de Química, Universidade Federal do Rio de Janeiro, Rio de Janeiro 21941-909, Brazil; dammakilyes@hotmail.fr; 3Embrapa Agroindústria de Alimentos, Empresa Brasileira de Pesquisa Agropecuária, Rio de Janeiro 23020-470, Brazil; ana.iraidy@embrapa.br; 4Embrapa Agroindústria Tropical, Empresa Brasileira de Pesquisa Agropecuária, Fortaleza 60511-110, Brazil; 5Laboratory of Inspection and Technology of Milk and Derivatives, Escola de Medicina Veterinária e Zootecnia, Universidade Federal da Bahia, Salvador 40170-110, Brazil; 6National Institute of Health Quality Control, Fundação Oswaldo Cruz, Rio de Janeiro 21040-900, Brazil

**Keywords:** food processing, non-thermal technologies, sonication, caprine milk, physicochemical stability, microbial viability

## Abstract

Despite goat milk having health benefits over cow milk, goat milk yogurt (GY) presents low consistency and viscosity, which reduces its overall acceptability by the consumer. Thus, new innovative methods can be an alternative to improve the quality of GY. Hence, this study aimed to investigate the effect of ultrasound (US) treatment with different sonication times on quality parameters of probiotic GY during refrigerated storage. US treatment was conducted at 20 KHz for 3, 6, and 9 min in yogurt. *Lactobacillus bulgaricus* and *Lactobacillus acidophilus* LA-5 were sensitive to US treatment, presenting a decrease in the yogurts stocked. This loss of viability led to reduced post-acidification due to smaller lactose metabolization in yogurt samples submitted to the US. Among tested treatments, the application of 6 min enhanced the apparent viscosity and consistency index of GY yogurts. In addition, this time also reduced tyramine and total biogenic amine (BAs) content. These findings suggest that 6 min of sonication is a promising way to improve the rheological properties and reduce the acidity and BAs content in GY. Further studies should be performed to optimize the US setting conditions to preserve the probiotic culture viability in yogurts.

## 1. Introduction

Goat milk has several advantages over cow milk, such as its higher digestibility of fat and proteins, higher vitamin A, vitamin B, and calcium contents [1,2]. In addition, goat milk has a hypoallergenic activity due to a lower level of α_s1_-casein than that of cow milk [3]. However, these characteristics from goat milk (fine fat globules, small casein micelles, and a low α_s1_-casein content) result in a low consistency and apparent viscosity in goat milk yogurt (GY) [4]. These factors also influence the rheological properties of the yogurt coagulum, which is semi-liquid. The weak gel structure, typical of GY, is detrimental to the product quality. Since this sensory attribute is the most relevant to the overall acceptability of yogurts, this contributes to GY rejection by consumers [5,6].

Many conventional methods have been used to improve the textural quality of GY, such as increasing solids in milk (adding fat, proteins, or sugars) as well as the addition of the stabilizer (pectin, starch, alginate, and gelatin). However, the suitable ratio between additives and goat milk for yogurt manufacture has not yet been achieved. This limitation can be attributed to this relation being very variable, depending on the type of additive and the technological process used [7]. These alternatives to improve GY texture may change nutritional food values [7] and bring unwanted flavors and textural attributes to the product. Fortification with these expensive dairy commodities also affects production costs [8]. In this context, reports in the literature have endeavored the development of innovative methods to improve the textural and rheological properties of GY [9,10].

Ultrasound (US) is defined as sound waves whose frequency exceeds the human ear (≈20 kHz). US technology has been an alternative to conventional food processing because it is safe and cheap [11]. Milk gels and yogurt produced from milk treated by the high-intensity US have shown enhanced firmness and viscosity and reduced syneresis in fermented cow milk, improving the physical properties of the product [12,13,14,15,16,17]. The changes in fermented cow milk by US are associated with whey protein denaturation. Once denatured, the whey proteins are more susceptible to association with casein and casein micelles, resulting in a stronger yogurt coagulum [17]. However, a negative consequence can be the biogenic amines (BAs) production, which has been neglected [18].

BAs are low-molecular-mass organic bases with an aliphatic, aromatic, or heterocyclic structure. They are produced from free amino acids decarboxylation. BAs represent a considerable toxicological risk in fermented dairy products [19] because free amino acids are naturally present in milk or are released from milk proteins by inherent proteolytic activities, as well as those from the starter and probiotic cultures or contaminating microorganisms. Tyramine is a vasoconstrictive amine that is predominant in fermented dairy products [20], including goat’s milk yogurts [21,22], due to the high tyrosine content in goat’s milk [23]. However, only aged cheeses, or those elaborated from raw milk, [24] such as “Jben” fresh cheese from raw goat’s milk [25], are typically described as containing clinically significant levels of tyramine per serving of food. In contrast, yogurts made from pasteurized milk and stocked in a suitable refrigeration temperature range (4 to 10 °C) are reported without a clinically significant content of tyramine per serving of food. A definition of a clinically significant level was related to the severity of elevation of the blood pressure raised from the consumption of tyramine naturally contained in food [24]. Regarding polyamines, their concentration (putrescine, cadaverine, spermidine, and spermine) in organs and tissues depends on their endogenous production and oxidation rates and their intake from foods [26]. Dietary polyamines may be harmful, neutral, or beneficial, depending on the specific polyamine involved and disease [27]. In this context, a positive or detrimental effect has been reported for yogurt consumption depending on the cancer type and bacterial strains used in the manufacture of yogurt; evidence supports the potential role of consumption of probiotic yogurt in the reduction of colorectal cancer risk in healthy adults [28,29]. In line, the consumption (100 g/day for 2 weeks) of *Bifidobacterium lactis* LKM512-containing probiotic yogurt reduced by 79.2%, while conventional yogurt decreased by 47.9% the gut mutagenicity level in healthy adults [30]. The consumption of yogurt with probiotic LKM512 (100 g/day for 2 weeks) also presented a protective effect in healthy elderly, significantly reducing the haptoglobin content in feces and gut mutagenicity level [31]. The antimutagenic effect on intestinal cells was attributed to the increased gut polyamines level, specially spermidine, by the consumption of probiotic yogurt, which led to the inhibition of inflammation [30,31]. On the other hand, limited suggestive evidence has shown that yogurt consumption increases the risk of prostate and kidney cancers in a dose-dependent manner in healthy adults, although this effect is not associated with the calcium content in yogurt [32,33,34]. Increased risk to develop these cancers types has been attributed to the polyamines content in dairy products [27], since they (putrescine and spermidine) are the other principal biogenic amines in yogurts [20,35], including goat’s milk yogurts [21,22]. Indeed, cadaverine, spermidine, and spermine present in conventional yogurts (1.9 to 2.8 µmol) were entirely absorbed by the human intestine, being yogurt suggested as a relevant source of polyamines for the organism [36]. Consistent with controversial findings on yogurt consumption and cancer risk, dietary spermidine is reported to have dual effects on cancers by targeting oncogenes, immunity, autophagy, or apoptosis [37]. In addition, although the intake of polyamines seems to decrease the risk of development of some cancer types and suppress tumorigenesis in cancer patients at the early stage, increased dietary spermidine appears to accelerate the growth of established tumors [37]. Thus, recent epidemiological studies have been shown that a polyamine-reduced diet can be beneficial to the quality of life of cancer patients [27]. In this context, evidence shows that increasing the sonication time leads to protein denaturation, which favors the proteolytic enzyme action in dairy products [38]; enhanced proteolytic activity is correlated to BAs accumulation in fermented dairy products [39], including goat’s milk yogurt [22].

Finally, the effects of the US on the viability and activity of starter and probiotic cultures have been controversial. The US conditions, bacterial species, and the matrix are parameters that influence an increase or decrease of bacterial viability and activity [3]. Nevertheless, the effects of the US on the main quality parameters of goat dairy products are still unknown and have been scarcely investigated in milk only [40,41]. This context requires an investigation into the effect of sonication conditions on GY quality. Thus, the major original hypothesis was that the optimized sonication time promotes desirable physical properties in GY. The additional hypothesis was that this time of sonication does not negatively affect the microbial quality and food safety of GY. Therefore, this study aimed to determine the sonication time effect (3, 6, and 9 min) on GY parameters (physicochemical and microbial) during refrigerated storage.

## 2. Results

### 2.1. Microbial Growth Evaluation of Freshly Prepared Goat Milk Yogurts

Table 1 shows the microbial count (log CFU g^−1^) on the first day of refrigerated storage (24 h after the yogurt manufacture). The values found here for the control (non-sonicated yogurt) were ≥7.44 log CFU g^−1^ both for the starter and the probiotic culture. However, the US treatments significantly reduced the viability when compared to the control: up to 22.53%, 41.4%, and 50.81% for *S. thermophilus*, *L. bulgaricus,* and *L. acidophilus* LA-5, respectively.

*S. thermophilus* was the most resistant species to sonication. It maintained scores ≥7.98 log CFU g^−1^ in all sonication times tested. On the other hand, *L. bulgaricus* and *L. acidophilus* LA-5 had their counts decreased to ≤5.01 log CFU g^−1^, even for the shortest sonication time (3 min). All bacterial species presented a decrease in cell viability when the US exposure time increased (Table 1). However, there was no significant difference (*p* > 0.05) between average and maximum sonication times (6 and 9 min).

### 2.2. Chemical Composition of Freshly Prepared Goat Milk Yogurt

Table 1 shows the results of the chemical composition of GY samples on the first day of storage. In general, moisture, minerals, fat, and protein content found here were similar between the control and the US treatments. Thus, the US had no influence (*p* > 0.05) on the chemical composition of yogurt samples.

### 2.3. Monosaccharides, Disaccharides, and Organic Acids Content in Goat Milk Yogurt during Storage

As shown in Table 2, lactose and lactic acid were the major disaccharide and organic acid, respectively, found in yogurt samples (both control and US treatments). The lactose content increased significantly with the sonication time, although in 28 days of storage, this tendency was not significant (Table 2). The sonication of 9 min (US9) resulted in the highest lactose accumulation compared to the control (non-sonicated yogurt). Therefore, US treatments reduced the lactose hydrolysis to glucose and galactose. Consistently, the galactose content decreased with US treatments (Figure 1B,F,J; Appendix A), although this trend was not significant on the first storage day (Table 2).

Glucose accumulated in the US3 and US6 treatments compared to the control (Table 2), indicating that exposure times of 3 and 6 min reduced the activity of glucose converting enzymes to organic acids. Consistently, 3 min of sonication significantly reduced the citric acid content. In contrast, the formic acid content decreased in all the samples exposed to the US than the control (Figure 1C,G,K). In addition, a significant and positive correlation was observed between the viability of dairy cultures exposed to treatments and the formic acid content (Figure 1C,G,K; Appendix A). Among the samples treated with ultrasound, US3 and US6 presented the highest galactose and formic acid content, exhibiting behavior more similar to the control (Table 2). Therefore, US3 and US6 showed a more active process of lactose hydrolysis than US9 treatment during storage.

Regarding storage time, lactose levels reduced only in the control and US6 in the two first weeks of storage when compared to fresh yogurt (1 day of storage) (Table 2), which indicates that significant hydrolysis of lactose occurred only in these samples. Glucose accumulated only in US3, which suggests a decrease in the activity of glucose-converting enzymes to organic acids for this sonication time. The citric acid significantly fluctuated during storage only in control (Table 2); thus, the production followed by consumption of the citric acid by LAB (lactic acid bacteria) metabolism just occurred in non-sonicated yogurt. The formic acid significantly fluctuated only in US3 and US9 treatments. The maintenance of the formic acid level in US6 and control indicates that the glucose conversion to formic acid remained constant in these samples. Consistently, there was an interaction between the storage period and sonication treatment for formic acid and glucose values (*p* < 0.05).

### 2.4. Evaluation of pH Values of Goat Milk Yogurt Samples during Storage

The pH values of yogurt samples during refrigerated storage are presented in Figure 2. The US treatments reduced the post-acidification of yogurt (*p* ˂ 0.05) in 28 days of storage, with the US9 treatment presenting the lowest acidity compared to the control. Moreover, a negative correlation (*p* ˂ 0.05) between the cell viability of treatments and pH was found (Figure 1A,E,I; Appendix A).

The pH results agreed with the reduced lactose hydrolytic activity and consequent low production of organic acids described here for samples submitted to ultrasound (Table 2; Figure 1). Among the US treatments, US3 and US6 presented the lowest pH values, which confirms that these treatments are the ones that most hydrolyzed lactose and produced formic acid (Figure 1). On the other hand, the storage time had an insignificant influence on the pH values (*p* > 0.05). These results indicate that the increased lactose hydrolysis during storage observed for the control and US6 (Table 2 and Figure 1) was insufficient to decrease pH values.

### 2.5. Biogenic Amine Content during Goat Milk Yogurt Sample Storage

The concentrations of total and individual biogenic amines (BAs) are shown in Table 3. Among BAs, tyramine predominated in all goat milk yogurt samples. The US processing did not affect the putrescine values (*p* > 0.05), except on the 14th day of storage, where the ultrasound slightly reduced its contents in all the treatments compared to the control. Additionally, the different sonication times did not present a significant influence on the putrescine content in the samples. Polyamines significantly reduced in US6 and US9 compared to the control. However, this reduction occurred over the whole storage period for spermidine, while the spermine content reduced only half of the storage time. Moreover, US6 and US9 did not differ from each other as to the spermidine content. The tyramine content decreased in all ultrasound treatments compared to the control (*p* < 0.05) in 14 and 28 days. Only on the first day of storage did the treatments not present any effect (*p* > 0.05). In addition, the tyramine reductions in 6 and 9 min of sonication were more dramatic than those in 3 min. The total BA content significantly reduced in all ultrasound treatments compared to the control until 14 days of storage. However, at the end of storage, the ultrasound did not influence the total BA content. In general, different sonication times did not affect the total BA content. In contrast, cadaverine increased in 6 and 9 min of sonication when compared to the control (*p* < 0.05) for the whole storage period (Table 3). Its content was higher in 6 min than in 9 min. Additionally, the BA concentration had a direct correlation with the viability of *S. thermophilus*, *L. bulgaricus*, and *L. acidophilus* by US treatments (*p* < 0.05) in yogurt samples (Figure 1; Appendix A).

The BA concentration presented a general increasing trend during storage compared to fresh yogurt (1 day of storage) in both the control and US treatments (Table 3). This increase was significant (*p* < 0.05) for cadaverine in NSU, US3, and US6 treatments. For putrescine, an increase was observed only in NSU from the 14th day of storage. Spermidine increased in all treatments. Furthermore, spermine increased in NSU, US3, and US6. Tyramine and the total BA content did not change in the NSU during the storage time, and they presented significance only in two treatments: US3 and US6 for tyramine, and US6 and US9 for total BAs. Consistently, there was no interaction (*p* > 0.05) between storage time and US treatments only for tyramine and total BAs.

### 2.6. Textural Properties during Goat Milk Yogurt Storage

The texture results are presented in Table 4. Firmness in US6 and US9 was slightly greater (36.59 and 34.70 g, respectively) than in NSU (34.16 g) in fresh yogurt. On the other hand, all US treatments resulted in slightly less firm samples (25.79 to 30.45 g) than the control (33.55 g) on the 14th day of storage, while they did not significantly differ at the end of storage. For consistency, US6 (384.48 g·s) presented a slightly higher value, while US3 and US9 (347.71 and 356.80 g·s, respectively) had slightly lower values when compared to the control (360.25 g·s) in fresh yogurt. However, all US treatments (254.59 to 309.66 g·s) presented lower consistency than the control (365.24 g·s) on the 14th day of storage, while they did not significantly differ at the end of storage. Thus, US treatments poorly contributed to the firmness and consistency of yogurt. Cohesiveness generally increased with firmness reduction for US treatments and control (Table 4).

The storage time reduced (*p* < 0.05) the firmness of all samples compared to fresh yogurt (1 day of storage). The same occurred for consistency, although this tendency had not been significant for US9 and the control. Coherently, the interaction between storage time and sonication treatments was significant (*p* < 0.05).

### 2.7. Rheological Behavior and Apparent Viscosity during Goat Milk Yogurt Samples Storage

Figure 3A1–A3 exhibits apparent viscosity as a function of shear rate over 28 days of storage in GY at 4 °C. Viscosity decreased with the increase in the shear rate during shearing, while shear stress increased as a function of shear rate for all samples (Figure 3B1–B3). All US treatments had a higher viscosity rate than the control (Figure 3A1–A3). US6 presented the highest viscosity value in fresh yogurt and at the end of storage. US3 showed the highest value on the 14th day, followed by US6 and US9. This indicates that overall, the middle time of sonication (6 min) was the most effective in improving product viscosity.

The rheological properties of yogurt samples—expressed as the consistency index (K) and flow behavior index (*n*)—are represented in Table 4. K values give an idea of the fluid viscosity; therefore, they were consistent with our finding for viscosity: K was only significantly higher than in the control for US6 in fresh yogurt (1.81 and 13.90 Pa s*^n^*, respectively) and at the end of storage (0.08 and 1.13 Pa s*^n^*, respectively). On the 14th day, all US treatments had a greater K than the control (0.04 Pa s*^n^*), but US3 and US6 (0.84 and 2.26 Pa s*^n^*, respectively) presented the highest values among US treatments. Thus, the middle time of sonication (6 min) was the most effective to improve the viscosity and consistency index of yogurt. The index *n* indicates the degree of deviation from the Newtonian flow (*n* = 1). All GY samples presented *n* < 1.0. The US6 treatment reduced the flow behavior index compared to the control in fresh yogurt, while it did not differ from the control on the 14th day of storage. However, on this day, the flow behavior index was higher in US3 and US9 than in NSU. Treatments did not differ at the end of storage (*p* > 0.05).

The storage time generally reduced the viscosity values for all samples compared to fresh yogurt (Figure 3A1–A3). These results are consistent with the reduction in the consistency index (K) observed in all samples compared to fresh yogurt during storage (Table 4). In contrast, the flow behavior index increased during storage in all samples compared to fresh yogurt (Table 4).

## 3. Discussion

In the current study, the count values for the starter and the probiotic cultures in the control sample (non-sonicated yogurt) were ≥7.44 log CFU g^−1^ (Table 1), which agree with those reported for viability in commercial dairy products (≥6 log CFU g^−1^) during cold storage [42]. However, the US treatment samples in the present work, unlike the control sample, cannot be characterized as probiotics because counts of probiotic *L. acidophilus* LA-5 were below the recommended minimum dose (6 log CFU g^−1^) to present the health benefits of this organism [43]. The susceptibility to the US varied according to the bacterial species studied (Table 1). Consistently, the sonication effect has been reported as a culture-specific treatment due to variations in cell wall thickness, composition, and cell size [13].

US effects on the viability and activity of the starter and probiotic cultures have been controversial because of the US setting conditions and particularities of the bacterial species and the matrix used in fermentation [3]. It may be explained by sonoporation, which increases the cell membrane permeability due to the US application [44]. Sonoporation can improve microbial growth since it facilitates the mass transfer of substrates or reagents across the cell membrane and the remotion of by-products of cellular metabolism. However, higher degrees of the US promote irreversible sonoporation, leading to the leakage of cellular content. Thus, the physical disruption and/or alteration of the cell membrane lipid bilayer cause lipid peroxidation and, eventually, cell death [45].

The lack of standardization in US operating conditions hinders comparisons between studies [45]. Nevertheless, for wave frequencies similar to those adopted herein (20 kHz), positive reports for the growth of starter and probiotic cultures were obtained for prolonged exposure time (up to 15 min), but with a power level (up to 150 W) and US wave amplitude (up to 25%) [46,47] lower than those employed herein (300 W and 67% of amplitude). Similarly, when the US treatment was performed before milk inoculation, even using a high-power level (up to 600 W) and time (up to 10 min), the starter and probiotic cultures presented a positive growth [48]. Activating inoculum in a yogurt starter culture using the US with a low power level (84 W) and short exposure time (150 s) before its introduction in milk improves cell viability [45]. A high power and wave amplitude (750 W and 100%, respectively) by prolonged time (10 min) reduces *L. acidophilus* count by 84% [49]. Thus, as investigated in this study, shorter exposure times with lower wave amplitude and power levels should be employed to prevent excessive sonoporation. Thus, the microbial quality can be maintained and accelerate fermentation during product elaboration. Consistently, lower exposure times to the US were reported to increase the survival percentage of LAB in milk and accelerate the fermentation process [50].

In general, the moisture, mineral, fat, and protein content found here (Table 1) was similar to that reported in the literature for goat milk yogurt [51]. US treatments do not influence the chemical composition of yogurt [47], which agrees with our findings. The literature has well established that the cavitation effect of the US only affects the particle size and distribution of yogurt compounds, not their concentration [40]. This fact justifies why the chemical composition of yogurt samples does not change after US treatments (Table 1).

Lactose and lactic acid are the major disaccharide and organic acid, respectively, in fermented dairy products [52,53], as observed herein (Table 2). However, the lactose values (40.16 to 44.99 mg g^−1^) reported for goat’s fermented milk [53] were slightly lower than those found by us (48 to 50 mg g^1^) in goat’s milk yogurts (Table 2). It can be justified by possible differences between both studies regarding the lactose content in the unfermented milk used to elaborate fermented dairy products. Coherently, the average content of lactose in goat’s milk has been reported to vary (44.3 to 59.7 mg/g) according to the breed and stages of lactation of the dairy goat [54]. In line, Bagnicka et al. [55] demonstrated that breed, parity, and litter size affect the content of lactose in the goat’s milk. LAB acidifies milk by lactose fermentation with organic acids production. Initially, the β-galactosidase hydrolyzes lactose to galactose and glucose [56]. Therefore, lactose accumulation (Table 2) accompanied by galactose reduction (Figure 1B,F,J; Appendix A) in our US-treated yogurt samples indicates a reduction in the activity of β-galactosidase released by LAB when the US was applied. Then, the glucose released from lactose is converted to pyruvate, which is metabolized by lactic dehydrogenase [17] or pyruvate-formate-lyase [57] to lactic acid or formic acid, respectively. Therefore, the highest glucose (Table 2) indicates that exposure times to the US of 3 and 6 min reduced the activity of enzymes converting glucose to organic acids. In contrast, there are reports that the US may improve the process of lactose hydrolysis by enhancing the membrane permeability of dairy cultures, allowing the release of intracellular β-galactosidase out from the cell. This release improves the hydrolyzing effect compared to the condition where β-galactosidase remains in the cells of dairy cultures [46].

Better lactose hydrolysis leads to the secretion of glucose and galactose and enhanced organic acid production [46]. However, when the US treatment reduces the viability of dairy cultures, as observed in this study (Table 2; Figure 1), it also reduces the production of organic acids in the cells by LAB. Organic acids are the main end products of lactic fermentation; therefore, the number of viable cells determines the amount of organic acid in the medium [58]. Lactose hydrolysis in our samples during storage compared to fresh yogurt may be explained by β-galactosidase released by LAB, which can remain active even at refrigerated storage temperature (0–5 °C) [59], as we observed in US6 and NSU (Table 2). Regarding organic acids, citric acid is the primary substrate for acetoin and diacetyl formation [52], which justifies its fluctuations observed in non-sonicated yogurt over storage in the present work (Table 2).

The pH values of yogurt samples during storage (Figure 2) coincided with those reported in the literature (4.1–4.6) for goat’s yogurt samples during cold storage [4,53]. During yogurt processing, the fermentation of lactose into organic acids by LAB lowers the pH value [49]; therefore, the most critical factor related to pH change during the dairy fermentation process is the LAB activity [38]. It explains the reduced post-acidification after US treatments (Figure 2) since they significantly reduced the dairy culture viability in the present work (Table 1). On the other hand, as reported by Jalilzadeh et al. [38], when the US stimulates the starter culture growth, it increases and accelerates acidity during fermentation due to the increased release of intracellular β-galactosidase from LAB cells [46]. The reduced post-acidification after US treatments (Figure 2) is technologically positive, because although sourness is an expected characteristic of yogurt, excessive post-acidification during storage may lower acceptance. In addition, post-acidification may affect textural quality and decrease the organoleptic quality of yogurt [56].

Consistently with our results (Table 3), tyramine has been reported as the BAs predominant in fermented dairy products [20]. The values found here for total and individual BA concentrations agree with those previously described for yogurt in cold storage [35]. The reduction in the total and individual BA content obtained for US-treated GY (Table 3) may be attributed to a reduction in the viability of the starter and probiotic cultures when the US was applied (Table 1). Coherently, in goat milk yogurt, the yogurt culture viability was previously considered as a significant variable in BA content [22]. Pintado et al. [60] reported strong and significant correlations between the concentration of BAs and viable microbial numbers in cheese. The lower the number of positive amino acid decarboxylase cells, the lower the number of amino acids converted to BAs [22], which justifies a lower BA production in the samples when the viability of dairy cultures is reduced. US6 and US9 had similar effects on the BA content (Table 3), which may be explained by the similar viability of *L. bulgaricus* and *L. acidophilus* LA-5 for these times in the present work (Table 1). On the other hand, the accentuated decrease in BAs with the sonication may be justified by the lower viability of *S. thermophilus*, *L. bulgaricus*, and *L. acidophilus* LA-5 (Table 1). In contrast, the cadaverine content increased in this study for US6 and US9 compared to the control (NSU). This effect may be associated with the viability of *S. thermophilus*, which has been reported to accumulate cadaverine, mostly due to lysine decarboxylase action [61]. Consistently, the count of *S. thermophilus* remained above the minimal level of the starter culture (7 log CFU g^−1^) [42] for all US treatments (Table 1). Regarding storage time, the BA content increased in all treatments compared to fresh yogurt (Table 3). This behavior is attributed to the amino acid decarboxylase enzyme, which remains active even at refrigerated storage (0–5 °C) [19].

Although there is no specific legislation regarding the BA content in dairy products, a consensus says it should not accumulate [20]. Therefore, the low total BA concentration observed in all GY samples (from 25.84 to 32.79 mg L^−1^; Table 3) indicates that the products are safe in this regard. Tyramine is the BA most frequently associated with BA-mediated dairy-borne intoxications. Moreover, this biogenic amine may be potentiated by putrescine and cadaverine [19]. Although there is no legislation about the occurrence of tyramine in foods, the European Food Safety Authority (EFSA) [62] proposed a no adverse effect levels (NOAEL) for tyramine, considering each meal intake during the day. NOAEL of 600 mg and 50 mg (per meal per person) for tyramine were established, respectively, for healthy individuals and those taking third-generation monoaminoxidase inhibitory (MAOI) drugs. Finally, NOAEL of 6 mg (per meal per person) for tyramine was proposed for individuals taking classical MAOI drugs. France is one of the countries that most consume yogurt in the world [63], with an average annual volume of 19.62 L per capita (from 2008 to 2018) [64,65]. Taking France as an example, the average content of tyramine in yogurt in the present study (Table 3) would lead to a daily tyramine intake of 0.97, 0.89, 0.89, and 0.85 mg per capita, respectively. Thus, none of our goat milk yogurt samples raised a health concern to tyramine content, since its level was below the NOAEL.

The effect of the US on the BA content in fermented dairy products was not found in the literature. Therefore, this study is the first report of this consequence in yogurt. Wójciak et al. [18] reported a reduction in the BA content (cadaverine, putrescine, and tyramine) by US treatment (40 kHz and acoustic power 480 W) in dry-fermented beef during ripening (93 days). Although they used US waves with higher energy and acoustic power values, their US treatments did not reduce the LAB viability, contrary to what was observed in the present study (Table 1). This difference may be justified by the influence of the matrix—where the US wave is propagated—on the viability of LAB culture [45].

Regarding the US effect compared to conventional technological strategies to improve the texture of yogurt, the addition of fiber-rich fruit pulp (viscous) was reported to increase by 4.29- and 1.70-fold the putrescine and spermine levels, respectively, in sheep’s milk yogurt [66]. For cow’s milk yogurt, the fruit addition increased putrescine (1.53 fold) and cadaverine (3.15 fold) [35]. On the other hand, the US treatment increased only cadaverine up to 1.44-fold in 6 min of sonication (Table 3). Regarding tyramine, fruit addition reduced 2.01- and 2.89-fold compared to control in sheep’s [66] and cow’s milk yogurt, respectively. For total BAs, the reduction was 1.19-fold in cow’s milk yogurt [35]. Our US treatments decreased tyramine and total BAs up to 1.14- and 1.17-fold, respectively, in 9 min of sonication (Table 3). Therefore, although fruit pulp led to a higher reduction in tyramine than US treatments, this effect was overlapped by an increase in other amines, so that the reduction in total BAs was similar in both treatments. Vieira et al. [35] described a similar behavior for cow milk yogurt with carbohydrate addition, where they observed an increase in cadaverine (5.03-fold) and a reduction in tyramine (2.60-fold), resulting in a reduction in total BAs by 1.19-fold, which is similar those reported here for US treatments (1.17-fold).

The data reported here for firmness (Table 4) were in line with Costa et al. [4], Park et al. [67], and Miocinovic et al. [68] for goat milk yogurts stored at 4 °C and unfortified with gums or protein isolate. Moreover, our results agree with Jalilzadeh et al. [38], who reported no difference in hardness between US-treated and control cheese samples during 60 days of ripening. The low ability of US treatments to improve the firmness and consistency in our study (Table 4) may be explained by an increase in the number of the particles, including fragments of gel, by a factor of approximately 2.5 in yogurt by US [14,69]. Thus, sonication can result in weaker gels than respective control yogurts [69,70]. In general, the firmness tended to reduce with ultrasound herein. However, this trend was not significant (*p* > 0.05) in most of the cases (Table 4). On the other hand, yogurt gels produced from milk treated by the high-intensity US have shown enhanced firmness [12,13,14,15,16,17]. This suggests that ultrasound application in milk instead of yogurt may be a better alternative to obtain a firmer product. In addition, particles in yogurt can re-aggregate during its storage [47]. The similar firmness and consistency between the control and US treatments at 28 days (Table 4) indicate that this re-aggregation occurred in all samples at the end of storage. The reduction in textural parameters during storage for all treatments compared to fresh yogurt (Table 4) agrees with the findings of Kamble and Kokate [71] and Vieira et al. [22]. They reported a decrease in firmness and consistency in cow’s yogurt in cold storage. *Lactobacillus acidophilus* and *Streptococcus thermophilus* have a proteolytic activity that hydrolyzes the protein network to small peptides and free amino acids during storage, resulting in significant texture reduction [72].

In general, the viscosity values (Figure 3A1–A3) for treatments from 14 days of storage, as well as for control during the whole storage period, were similar to those reported by Costa et al. [4] and Gursoy et al. [47] for yogurts during refrigerated storage. On the other hand, values for treatments on the first day of storage, US3 on the 14th day, and US6 after 28 days of storage were similar to those reported for goat’s yogurt fortified with solids or whey proteins [73]. In addition, the behavior of yogurts during storage (Figure 3B1–B3) highlighted the rheological profile of these samples as a non-Newtonian liquid with pseudoplastic behavior, which is typical of yogurt [74]. The results indicate that US treatments did not considerably affect the flow behavior of samples, which remained typical of yogurt. Enhanced viscosity by US treatments, as observed herein (Figure 3A1–A3), was previously reported by Gursoy et al. [47]. They found an increase in apparent viscosity due to sonication (125 to 150 W for 15 min) in cow’s milk yogurt in 10 days of storage. As the US increases the particle number, including fragments of gel [14,69], the US after inoculation improved the viscosity in the finished yogurt [75], as also observed herein (Figure 3).

High values of determination coefficient (ranging from 0.911 ± 0.031 to 0.994 ± 0.001) indicate that the power-law model was appropriate to determine the rheological properties of GY. The values of consistency index (K) and flow behavior index (*n*) (Table 4) were in line with those reported by Ramírez-Sucre and Vélez-Ruiz [76] and Keogh and O’Kennedy [77] for yogurt. Similar to our results (Table 4), Barukčić et al. [48] reported an increased consistency index in sonicated milk whey samples (480 W for 10 min) compared to control. The increase in K values by US may be attributed to the reasons previously discussed here, which led to an increase in apparent viscosity. The middle time of sonication (6 min) was the most effective to improve viscosity (Figure 3) and increase the consistency index (Table 4). This indicates that the shorter time used (3 min) may not have been sufficient to dissociate gel structure in smaller particles [75] significantly. On the other hand, the longer time of sonication (9 min) may have disrupted the protein gel and affected the interaction between proteins during storage [48].

The flow behavior index (*n*) indicates the degree of deviation from the Newtonian behavior (*n* = 1). If *n* is greater than 1, the fluid is classified as dilatant, presenting shear-thickening; if *n* is between zero and 1, the fluid is classified as pseudoplastic or non-Newtonian, exhibiting shear-thinning. All samples were considered non-Newtonian liquids with pseudoplastic behavior, as expected for yogurt rheology [74]. This reinforces our finding for viscosity behavior (Figure 3): the US treatment did not change the type of flow of samples. Similar behavior was reported for milk whey samples submitted to the US [48]. The decrease in the flow index by US6 reinforces the greater efficiency of this time of sonication to fragment the protein network, which increases the particle concentration in the product, leading to a reduction in *n* [76].

The reduction in the viscosity and consistency index values in all samples compared to fresh yogurt during storage (Table 4) agree with the findings of Kamble and Kokate [71], who reported a decrease in apparent viscosity in cow’s yogurt in cold storage. The reduction in these parameters may be attributed to the growth and release of proteolytic enzymes by starter cultures in yogurt during storage, which breaks down the gel structure to small peptides and free amino acids [72]. Finally, the increased flow behavior index in samples compared to the fresh yogurt (Table 4) may be related to the decrease of consistency index values during storage, since both rheological parameters present an inverse relation in pseudoplastic fluids [78].

## 4. Materials and Methods

### 4.1. Reagents and Chemicals

Standards of biogenic amines (tyramine, putrescine, cadaverine, spermine, and spermidine), mono- and disaccharides (lactose, glucose, and galactose), and organic acids (formic, citric, and lactic acids), all ≥ 98%, were purchased from Sigma-Aldrich (St. Louis, MO, USA). Acetonitrile (HPLC grade) and all analytical-grade chemicals were obtained from Tedia (São Paulo, Brazil). A Millipore Milli-Q water system was also used (Millipore, Bedford, MA, USA).

### 4.2. Preparation of Goat Milk Yogurt Samples

The UHT-processed whole goat milk (Caprilat^®^, Paraná, Brazil) used in the manufacture of the yogurt samples presented pH = 6.70 ± 0.20. In addition, the chemical composition of milk was of 3.5 ± 0.15, 3.0 ± 0.17, 0.70 ± 0.12, and 88.5 ± 1.0 g/Kg for fat, protein, ash, and moisture, respectively. The yogurt samples processing was performed by Costa et al. [4]. Thermophilic yogurt cultures (1% (*v*/*v*); YF-L903; Chr. Hansen, Valinhos, Brazil) and a *Lactobacillus acidophilus* probiotic culture (5% (*v*/*v*); LA-5^®®^; Chr. Hansen, Valinhos, Brazil) were used to inoculate whole goat milk (Caprilat^®^, Paraná, Brazil). The milk samples were incubated in a thermosetting incubator (Heratherm, Thermo Scientific™, Brazil) at 43 ± 2 °C. After 4 h of fermentation, the final pH (4.4) was reached, as described in the official method [79]. Then, fermentation was stopped by refrigerating the yogurts at 4 ± 1 °C. After 2 h of refrigeration, the yogurt samples at 4 ± 2 °C were subjected to the ultrasound treatments.

### 4.3. Ultrasound (US) Treatments

After fermentation, yogurt samples (150 mL) were subjected to US waves of 67% of amplitude in an ultrasonic bath at a frequency of 20 kHz (UIP1000hdT, Hielscher Ultrasonics GmbH, Teltow, Germany). The sonotrodes (18-mm diameter tip; Sonotrode BS2d18, Hielscher Ultrasonics GmbH, Teltow, Germany) generated a total power of 300 W, which corresponds to the US intensity 2 W/mL of material. Two batches of US treatment were performed.

The US treatment was carried out in an ice bath to minimize the extent of lipid oxidation; the temperature was kept at 8 ± 2 °C. To study the effect of time, different US treatment durations were tested, as follows: non-treated sample (NSU), US treatment for 3 min (US3), US treatment for 6 min (US6), and US treatment for 9 min (US9).

The ultrasound treatment times were chosen based on those reported in the literature for cow’s milk yogurt sonicated before [15,16] or after inoculum of starter cultures [13,14,46,58,75], as well as for unfermented cow and goat milk [12,40].

### 4.4. Microbial Analysis

The microbial count was performed in triplicate using a spiral plater (model Eddy Jet 2, IUL Instruments, Barcelona, Spain) on the first day of storage. The *Streptococcus thermophilus* count was performed using M17 agar (Difco Laboratories, Detroit, MI, USA), enriched with lactose and incubated under aerobic conditions at 37 ± 1 °C for 48 h; the count of *Lactobacillus delbrueckii* ssp. *bulgaricus* was performed using de Man, Rogosa, and Sharpe agar (MRS, Difco Laboratories, Detroit, MI, USA) with pH 5.4 after incubation under an anaerobic condition at 37 ± 1 °C for 72 h [42]. Following Costa et al. [4], the *L. acidophilus* LA-5 enumeration was performed in MRS agar (Difco Laboratories, Detroit, MI, USA) supplemented with 0.15% (*w*/*v*) bile salts incubated under microaerophilic conditions at 37 ± 1 °C for 48 h. The mean of the bacterial counts was calculated, in experimental and analytical triplicate, and expressed as log CFU g^−1^ in fresh yogurt (1st day of storage at 4 °C).

### 4.5. Chemical Characterization of Goat Milk Yogurt

After the ultrasound treatments, the samples were analyzed for chemical composition in fresh yogurt (1st day of storage at 4 °C) and for other chemical parameters at fourteen-day intervals (1, 14, and 28 days) during storage at 4 °C. The assays were performed in experimental and analytical triplicate.

#### 4.5.1. Chemical Composition

The mineral content (g/Kg) of yogurt samples was determined by sample incineration in an open inert vessel and destruction of the organic content by thermal decomposition at 550 °C within 2 h using a muffle furnace (FO100 Yamato, Tokyo, Japan), as described in the official method 900.02 [79]. The protein amount (g/Kg) was analyzed according to the Kjeldahl method, based on the nitrogen content of the yogurt samples and then expressed as protein content by multiplying the nitrogen content determined by 6.38, as described in the official method 992.23 [79]. The fat content (g/Kg) was analyzed using the Gerber method described in the official method 920.39C [79]. The moisture content was analyzed using infrared radiation drying (model LJ16, Mettler Toledo^®®^, Barueri, SP, Brazil) following the official method and expressed as g/Kg [79].

#### 4.5.2. Monosaccharides, Disaccharides, and Organic Acids Quantification by HPLC-DAD-RID

The mono- and disaccharides as well as the organic acids were extracted and quantified by high-performance liquid chromatography with a diode array detector and a refractive index detector (HPLC-DAD-RID), using a method previously described and validated by our research group [53]. Briefly, monosaccharides, disaccharides, and organic acids were extracted from yogurt samples (1 g) through homogenization with 5 mL of 45 mmol L^−1^ H_2_SO_4_ for 30 min in a shaker (TS-2000 A VDRL shaker, Biomixer^®^, São Paulo, Brazil) at 240 rpm following another 1 min in the vortex. The homogenates were centrifuged at 5500× *g* for 20 min at 4 °C (Sorvall ST16R, Thermo Scientific, São Paulo, Brazil). The supernatant was initially filtered through Whatman No. 1 filter paper and then passed through a 0.45 µm pore size membrane (PVDF, Millipore, Brazil) filter. After, the injection volume of 20 μL of monosaccharides, disaccharides, and organic acids was separated on an Aminex HPX-87H column (300 × 7.8 mm, 9 µm particle size, 8% cross-linkage and pH range of 1–3; Bio-Rad, Hercules, CA, USA), utilizing a 3 mM sulfuric acid aqueous mobile phase (pH 2.35) under isocratic conditions at a flow rate of 0.5 mL min^−1^. The column temperature was maintained at 60 °C. The wavelength for organic acid detection was set at 210 nm. The chromatographic system consisted of a LC-20AT pump integrated with CBM-20A controller and SPD-M20A diode array detector in-line with RID-10A refractive index serial detector (Shimadzu, Kyoto, Japan). The compounds were identified by retention times and by spiking the suspect analyte to the sample, and the concentrations (mg g^−1^) were determined by interpolation in standard external curves (0.0–60 mg g^−1^, r^2^ ≥ 0.995) using a LC Solution software version 2.1.

#### 4.5.3. Biogenic Amine Quantification by RP-HPLC-DAD

The extraction and quantification of biogenic amines (BAs) were performed by reverse-phase HPLC with a diode array detector (RP-HPLC-DAD). The methods of extraction and derivatization and the chromatographic conditions were previously described and validated following the US-FDA guidelines, which were considered suitable for yogurt quality control [35]. Briefly, BAs extraction from yogurt samples was performed with HClO_4_ (0.6 M) in a shaker (TS-2000 A VDRL shaker, Biomixer^®^, São Paulo, Brazil) for 30 min. After centrifugation at 5500× *g* for 30 min at 4 °C (Sorvall ST16R, Thermo Scientific, São Paulo, Brazil), the supernatant filtered through Whatman No. 1 filter paper was alkalinized with NaOH (2 M) for the precipitation of small peptides. After centrifugation (5500× *g* for 15 min at 4 °C), BAs derivatization in the supernatant was performed using benzoyl chloride. Then, twice the extraction of BAs derivatives with diethyl ether was performed, followed by evaporation under N_2_ and resuspension with acetonitrile. The chromatographic separation of 50 μL of the derivatized sample was achieved in a Kromasil^®^ C18 column (250 × 4.6 mm i.d., 5-μm particle size). A mobile gradient phase with a flow rate of 0.6 mL min^−1^ composed of ultrapure water (A) and acetonitrile (B) was used. Detection was performed in UV at 254 nm, and the column temperature was maintained with a thermostat at 40 °C. The HPLC system (Shimadzu, Kyoto, Japan) consisted of an LC-20AT pump, SPD-M20A diode-array detector, CTO-20A oven, and SIL-20AC autosampler, all of which were connected to a CBM-20A controller. The BAs were identified by retention times and by spiking the samples with the suspected amine and quantified by interpolating peak area in standard external curves (1–50 mg L^−1^, r^2^ ≥ 0.980) using LC Solution software. The data were expressed as mg L^−1^.

### 4.6. Physical Characterization of Goat Milk Yogurt

After ultrasound treatments, the samples were analyzed in experimental and analytical triplicate, for physical parameters at fourteen-day intervals (1, 14, and 28 days) during storage at 4 °C.

#### 4.6.1. pH Determination

The pH of each sample was measured with a digital pH-meter (ISTEK, Model 720P, Guro-dong, Guro-gu, Korea) equipped with a glass electrode, which was inserted directly into the yogurt. Before the measurement, the electrode was calibrated with standard buffer solutions of pH 4.00 and 7.00 [79].

#### 4.6.2. Textural Properties of Goat Milk Yogurt

Firmness (g), consistency (g·s), and cohesiveness (g) were measured according to Costa et al. [4], using a texture analyzer (TA-XT.Plus, Stable Micro Systems Ltd., Surrey, UK) equipped with a 49.0 N load cell. The back extrusion cell plunger was 3.6 cm in diameter and was placed at 20 mm above the sample surface. The test cell penetrated 2 cm into the sample at 4 °C.

#### 4.6.3. Rheological Behavior and Apparent Viscosity of Goat Milk Yogurt

The gels of yogurt samples were broken by stirring before rheological measurements, which were performed using a concentric cylinder Brookfield viscometer (LVDVIII) and a spindle SC4-34 (Brookfield Engineering Laboratories, MA, USA). For each measurement, 9 mL of yogurt were carefully deposited in the sample cup (11 mL). For all samples, the filled sample cup and the spindle were temperature-equilibrated for about 15 min. The flow curves of samples were determined at speeds between 20 and 250 rpm (20, 40, 60, 80, 100, 120, 150, 180, 200, and 250); their corresponding shear rates (γ̇) and shear stresses (σ) were computed from relations given by the instrument manufacturer and then recorded. All characterizations were performed three times for each sample.

The experimental data were fitted to the Power law model [80] as in Equation (1):Power law model: σ = K (γ̇) ^*n*^(1)
where σ is the shear stress (Pa), γ̇ is the shear rate (s^−1^), K is the consistency index (Pa·s^*n*^), and *n* is the flow behavior index (dimensionless).

The Wingather program (Brookfield Engineering Laboratories Inc., Stoughton, MA, USA) was used to collect data and calculate apparent viscosity. Viscosity values in the upward viscosity/shear rate curves at a shear rate between 4 and 25 s^−1^ (equivalent to rotation from 20 rpm to 250 rpm) were reported as the apparent viscosity of yogurt samples and expressed in mPa·s.

### 4.7. Internal Validation of Predictive Models

The performance of a predictive model is overestimated (optimism) when based only on the sample used to construct the model. Therefore, internal validation methods aim to provide an accurate estimate of model performance in new samples [81].

The estimate of Harrell’s optimism was calculated according to Equation (2) [82] and the coefficient of determination of the original model after validation (Equation (3)).
(2)$$o=\frac{\sum_{m=1}^{M}o^{\left(m\right)}}{M}$$
(3)$$R_{v}^{2}=R_{app}^{2}-o$$
where for each bootstrap sample with replacement (*m* = 1, …, *M*), *R*^2^*_boot_*^(*m*)^ = bootstrap coefficient of determination obtained from the fitted model to the bootstrap dataset; *R*^2^*_orig_*^(*m*)^ = original coefficient of determination obtained by applying the fitted model from the bootstrap dataset to the original dataset; *o* = optimism of the original model; $o^{\left(m\right)}=R_{boot^{\left(m\right)}}^{2}-R_{orig^{\left(m\right)}}^{2}$; *M* = number of bootstrap datasets; *R*^2^*_v_*: coefficient of determination of the original model after validation; *R*^2^*_app_* = apparent coefficient of determination obtained from fitted model to original data.

### 4.8. Statistical Analysis

All analyses were performed in experimental and analytical triplicate, and the results were expressed as mean ± standard deviation (SD). Significance tests were conducted using one-way analysis of variance (ANOVA) at a 0.05 significance level for the variables of chemical composition and bacterial counts. For the other variables, the two-way ANOVA analysis was applied (*p* ˂ 0.05). When a significant F was observed, differences between means were evaluated by Tukey’s multiple comparison tests, and two-side *p*-values < 0.05 were considered statistically significant. The correlation between variables was evaluated using Pearson’s correlation test with a significance level of 0.05. Next, the models were internally validated through the bootstrap method (confidence interval = 95%; number of simulations = 1000; size of bootstrap samples = size of original sample; and number of bootstrap samples = 200) [82]. The statistical analyses were performed using the XLSTAT software (version 2013.2.03; Addinsoft, Paris, France).

## 5. Conclusions

US technology has been proposed as an alternative to conventional food processing to improve the physical properties of dairy products. The US treatment in goat milk yogurt may improve its rheological properties depending on the exposure time. The exposure to the US for 6 min considerably enhanced the apparent viscosity and the consistency index of yogurt samples. Therefore, US processing for 6 min may be potentially used by the dairy industry to improve the physical properties of stored goat yogurt while reducing the BA content and post-acidification. However, further studies should be performed, since the direct application of US in yogurt interfered with the viability of probiotic bacteria.

## Figures and Tables

**Figure 1 molecules-25-04638-f001:**
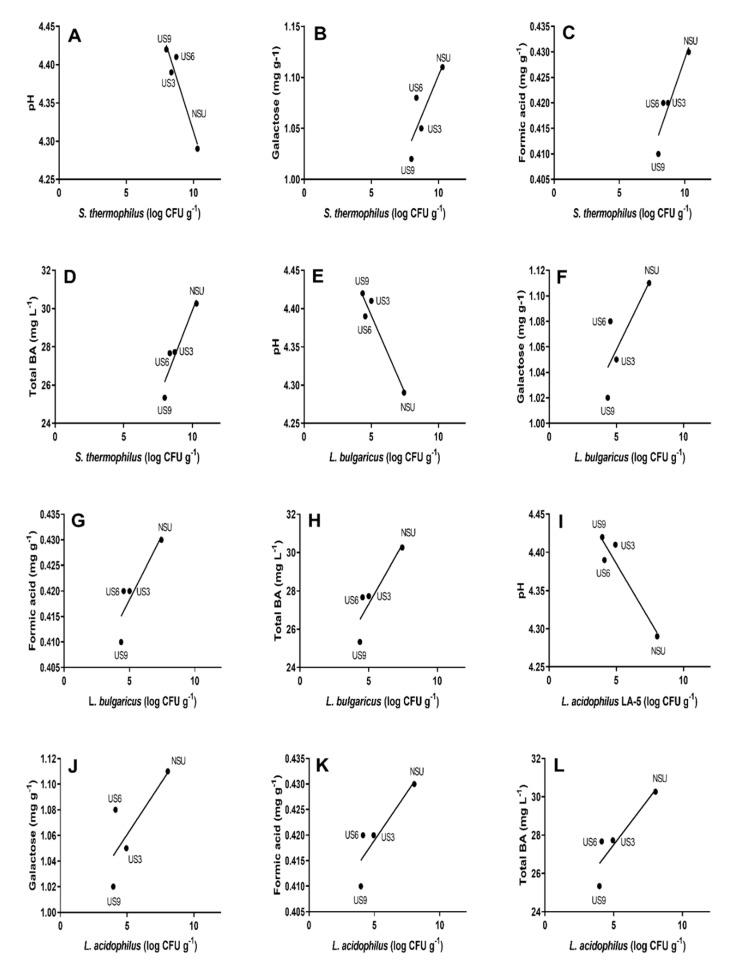
Significant correlations (*p* ˂ 0.05) were internally validated by the bootstrap method between physicochemical and microbial parameters for goat milk yogurt stored at 4 °C. (**A**–**D**) *Streptococcus thermophilus*; (**E**–**H**) *Lactobacillus delbrueckii* ssp. *bulgaricus*; (**I**–**L**) *Lactobacillus acidophilus* LA-5. NSU, non-sonicated goat milk yogurt; US3, goat milk yogurt sonicated for 3 min; US6, goat milk yogurt sonicated for 6 min; US9, goat milk yogurt sonicated for 9 min; total BA, total biogenic amines.

**Figure 2 molecules-25-04638-f002:**
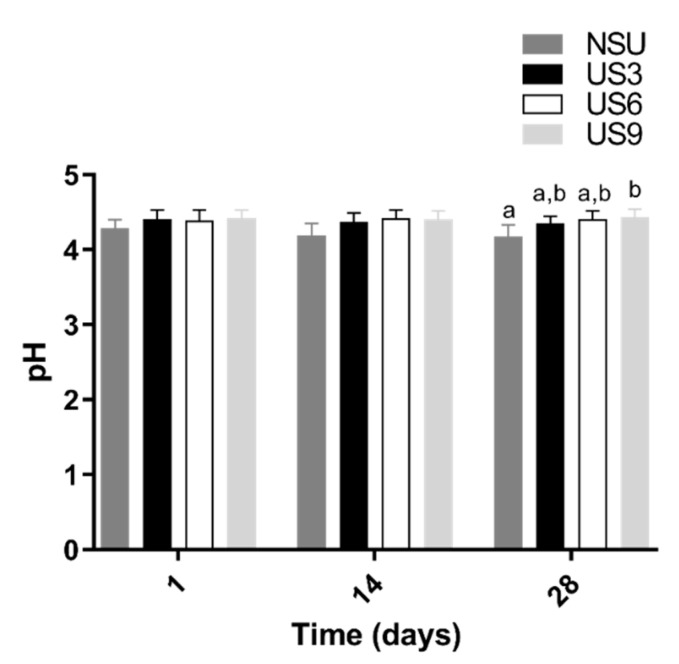
Changes in pH values in goat milk yogurt samples during storage at 4 °C. a–b Different lowercase letters indicate statistical differences between ultrasound treatments in goat milk yogurt samples during 28 days of storage (*p* < 0.05). NSU, non-sonicated goat milk yogurt; US3, goat milk yogurt sonicated for 3 min; US6, goat milk yogurt sonicated for 6 min; US9, goat milk yogurt sonicated for 9 min.

**Figure 3 molecules-25-04638-f003:**
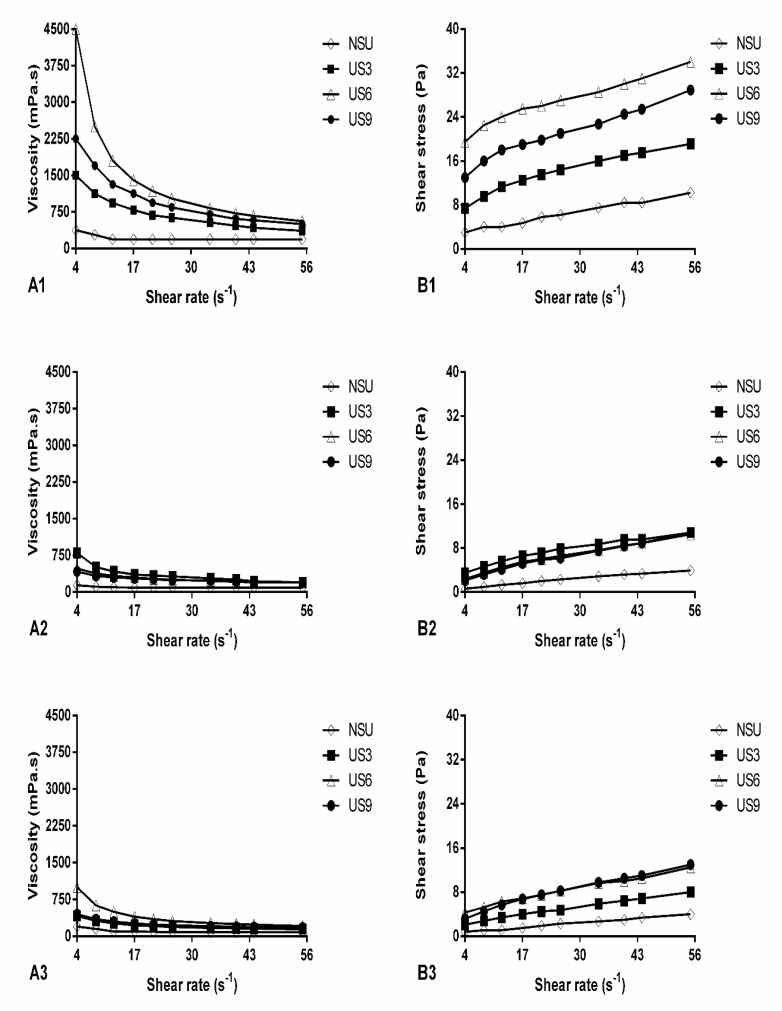
Flow behavior of goat milk yogurt samples during storage time at 4 °C. (**A1**–**A3**) apparent viscosity vs. shear rate; (**B1**–**B3**) Flow curves; (**A1**,**B1**) day 1 of storage; (**A2**,**B2**) day 14 of storage; (**A3**,**B3**) day 28 of storage. NSU, non-ultrasonicated goat milk yogurt; US3, goat milk yogurt sonicated for 3 min; US6, goat milk yogurt sonicated for 6 min; US9, goat milk yogurt sonicated for 9 min.

**Table 1 molecules-25-04638-t001:** Microbial counts and chemical composition in goat milk yogurts were measured on the first day of storage at 4 °C.

Parameter ^1^	Treatment ^2^			
	NSU	US3	US6	US9
*Lactobacillus bulgaricus*	7.44 ± 0.15 ^c^	5.01 ± 0.24 ^b^	4.56 ± 0.32 ^a^	4.36 ± 0.19 ^a^
*Streptococcus thermophilus*	10.30 ± 0.34 ^c^	8.72 ± 0.15 ^b^	8.36 ± 0.34 ^ab^	7.98 ± 0.38 ^a^
*Lactobacillus acidophilus LA-5*	8.05 ± 0.03 ^c^	4.94 ± 0.33 ^b^	4.13 ± 0.56 ^a^	3.96 ± 0.17 ^a^
Ash	0.70 ± 0.14 ^a^	0.80 ± 0.08 ^a^	0.77 ± 0.10 ^a^	0.72 ± 0.10 ^a^
Fat	3.33 ± 0.15 ^a^	3.30 ± 0.26 ^a^	3.47 ± 0.15 ^a^	3.30 ± 0.17 ^a^
Moisture	87.33 ± 0.74 ^a^	87.61 ± 3.23 ^a^	87.80 ± 0.44 ^a^	87.80 ± 0.01 ^a^
Protein	0.70 ± 0.11 ^a^	0.61 ± 0.08 ^a^	0.61 ± 0.16 ^a^	0.64 ± 0.1 2^a^

^a–c^ Different lowercase letters indicate statistical differences between ultrasound treatments of goat’s milk yogurts on the first day of storage (*p* < 0.05). ^1^ Bacterial counts were expressed as log CFU g*^−^*^1^; Chemical composition was expressed as g/Kg. ^2^ NSU, non-sonicated goat’s milk yogurt; US3, goat’s milk yogurt sonicated for 3 min; US6, goat’s milk yogurt sonicated for 6 min; US9, goat’s milk yogurt sonicated for 9 min. Mean ± standard deviation from triplicate determinations.

**Table 2 molecules-25-04638-t002:** Values of monosaccharides, disaccharides, and organic acids in goat milk yogurt samples measured during 28 days of storage at 4 °C.

Parameter ^1^	Treatment ^2^	Storage Time (Day)		
		1	14	28
Lactose	NSU	50.40 ± 1.59 ^ab,B^	48.05 ± 2.10 ^a,A^	48.60 ± 2.06 ^AB^
	US3	49.92 ± 2.54 ^a^	49.44 ± 2.71 ^ab^	49.47 ± 2.19
	US6	51.84 ± 1.02 ^ab,B^	49.60 ± 2.50 ^ab,A^	51.01 ± 0.74 ^AB^
	US9	52.62 ± 1.67 ^b^	51.91 ± 2.49 ^b^	51.25 ± 3.72
Glucose	NSU	0.10 ± 0.02 ^ab^	0.10 ± 0.01 ^a^	0.09 ± 0.01 ^a^
	US3	0.12 ± 0.03 ^bc,A^	0.15 ± 0.02 ^b,AB^	0.16 ± 0.02 ^c,B^
	US6	0.14 ± 0.03 ^c^	0.16 ± 0.04 ^b^	0.14 ± 0.01 ^b^
	US9	0.08 ± 0.01 ^a^	0.08 ± 0.00 ^a^	0.08 ± 0.00 ^a^
Galactose	NSU	1.11 ± 0.08	1.12 ± 0.06 ^b^	1.11 ± 0.09 ^b^
	US3	1.05 ± 0.06	1.10 ± 0.06 ^ab^	1.08 ± 0.05 ^b^
	US6	1.08 ± 0.03	1.07 ± 0.05 ^ab^	1.11 ± 0.01 ^b^
	US9	1.02 ± 0.12	1.01 ± 0.10 ^a^	0.99 ± 0.07 ^a^
Citric acid	NSU	0.13 ± 0.01 ^b,B^	0.11 ± 0.01 ^ab,A^	0.12 ± 0.01 ^b,B^
	US3	0.11 ± 0.01 ^a^	0.10 ± 0.01 ^a^	0.11 ± 0.01 ^a^
	US6	0.13 ± 0.01 ^b^	0.11 ± 0.03 ^ab^	0.13 ± 0.00 ^b^
	US9	0.13 ± 0.01 ^b^	0.13 ± 0.00 ^b^	0.13 ± 0.01 ^b^
Lactic acid	NSU	1.37 ± 0.08 ^ab^	1.37 ± 0.07	1.33 ± 0.14
	US3	1.26 ± 0.04 ^a^	1.30 ± 0.04	1.26 ± 0.04
	US6	1.39 ± 0.15 ^ab^	1.28 ± 0.12	1.32 ± 0.02
	US9	1.46 ± 0.19 ^b^	1.37 ± 0.14	1.32 ± 0.11
Formic acid	NSU	0.43 ± 0.01 ^b^	0.42 ± 0.01 ^c^	0.43 ± 0.01 ^b^
	US3	0.42 ± 0.00 ^ab,AB^	0.41 ± 0.01 ^bc,A^	0.42 ± 0.01 ^ab,B^
	US6	0.42 ± 0.00 ^ab^	0.41 ± 0.01 ^ab^	0.39 ± 0.05 ^a^
	US9	0.41 ± 0.01 ^a,B^	0.40 ± 0.01 ^a,A^	0.42 ± 0.00 ^ab,B^

^a–c^ Different lowercase letters indicate statistical differences between ultrasound treatments in goat’s milk yogurt samples during 28 days of storage (*p* < 0.05). A–B Different uppercase letters indicate statistical differences between days of storage in goat’s milk yogurt samples (*p* < 0.05). ^1^ Monosaccharides, disaccharides, and organic acids were expressed as mg g^−1^. ^2^ NSU, non-sonicated goat’s milk yogurt; US3, goat’s milk yogurt sonicated for 3 min; US6, goat’s milk yogurt sonicated for 6 min; US9, goat’s milk yogurt sonicated for 9 min.

**Table 3 molecules-25-04638-t003:** Biogenic amine concentration in goat milk yogurt samples was measured during 28 days of storage at 4 °C.

Biogenic Amines (mg L^−1^)	Treatment	Storage Time (day)		
		1	14	28
Cadaverine	NSU	2.80 ± 0.44 ^a,A^	2.80 ± 0.44 ^a,A^	3.90 ± 0.72 ^ab,B^
	US3	2.61 ± 0.53 ^a,A^	2.69 ± 0.57 ^a,A^	2.93 ± 0.72 ^a,A^
	US6	3.80 ± 0.92 ^b,A^	4.54 ± 0.90 ^b,AB^	5.47 ± 1.42 ^c,B^
	US9	3.01 ± 0.85 ^ab,A^	5.49 ± 1.27 ^b,B^	5.21 ± 1.22 ^bc,B^
Putrescine	NSU	2.56 ± 0.76 ^a,A^	3.83 ± 0.67 ^b,B^	4.12 ± 0.96 ^a,B^
	US3	2.80 ± 0.47 ^a,A^	2.91 ± 0.68 ^a,A^	3.16 ± 0.87 ^a,A^
	US6	2.79 ± 0.51 ^a,A^	2.88 ± 0.54 ^a,A^	3.34 ± 0.93 ^a,A^
	US9	2.61 ± 0.71 ^a,A^	2.77 ± 0.59 ^a,A^	3.16 ± 0.54 ^a,A^
Spermidine	NSU	2.34 ± 0.64 ^b,A^	4.53 ± 1.12 ^b,B^	5.87 ± 1.19 ^c,C^
	US3	2.32 ± 0.66 ^b,A^	2.24 ± 0.55 ^a,A^	5.04 ± 0.84 ^bc,B^
	US6	1.54 ± 0.36 ^a,A^	1.88 ± 0.47 ^a,A^	4.21 ± 0.90 ^ab,B^
	US9	1.29 ± 0.36 ^a,A^	1.55 ± 0.42 ^a,A^	3.39 ± 0.91 ^a,B^
Spermine	NSU	1.95 ± 0.33 ^a,A^	3.00 ± 0.75 ^c,B^	3.23 ± 0.97 ^b,B^
	US3	1.69 ± 0.20 ^a,A^	2.48 ± 0.65 ^bc,B^	2.95 ± 0.52 ^b,B^
	US6	1.78 ± 0.21 ^a,A^	1.52 ± 0.09 ^a,A^	2.49 ± 0.70 ^ab,B^
	US9	1.85 ± 0.29 ^a,A^	1.82 ± 0.40 ^ab,A^	1.93 ± 0.30 ^a,A^
Tyramine	NSU	16.50 ± 4.76 ^a,A^	18.35 ± 1.12 ^b,A^	19.34 ± 1.12 ^b,A^
	US3	14.33 ± 2.24 ^a,A^	17.30 ± 0.93 ^ab,B^	18.20 ± 1.67 ^ab,B^
	US6	16.00 ± 1.13 ^a,A^	16.24 ± 1.16 ^a,AB^	17.76 ± 1.65 ^ab,B^
	US9	14.93 ± 1.36 ^a,A^	15.96 ± 2.74 ^a,A^	16.58 ± 0.99 ^a,A^
Total concentration	NSU	30.26 ± 1.64 ^a,A^	32.79 ± 2.80 ^a,A^	32.06 ± 6.10 ^a,A^
	US3	27.73 ± 1.51 ^b,A^	27.87 ± 2.03 ^b,A^	28.05 ± 1.89 ^a,A^
	US6	27.67 ± 1.74 ^b,A^	27.05 ± 2.26 ^b,A^	31.51 ± 3.93 ^a,B^
	US9	25.34 ± 1.27 ^c,A^	27.58 ± 1.85 ^b,B^	28.61 ± 1.70 ^a,B^

a–c Different lowercase letters indicate statistical differences between ultrasound treatments in goat milk yogurt samples during 28 days of storage (*p* < 0.05). A–B Different uppercase letters indicate statistical differences between days of storage of goat milk yogurt samples (*p* < 0.05). NSU, non-sonicated goat milk yogurt; US3, goat milk yogurt sonicated for 3 min; US6, goat milk yogurt sonicated for 6 min; US9, goat milk yogurt sonicated for 9 min. Mean ± standard deviation from triplicate determinations.

**Table 4 molecules-25-04638-t004:** Texture profile analysis and rheological characteristics of goat milk yogurt samples measured during 28 days of storage at 4 °C.

Parameter ^1^	Treatment ^2^	Storage Time (Day)		
		1	14	28
F (g)	NSU	34.16 ± 0.41 ^a,B^	33.55 ± 0.26 ^c,A^	33.89 ± 0.27 ^a,AB^
	US3	33.56 ± 2.07 ^a,B^	27.26 ± 1.76 ^a,A^	32.22 ± 3.10 ^a,B^
	US6	36.59 ± 0.78 ^b,C^	25.79 ± 1.61 ^a,A^	31.45 ± 2.94 ^a,B^
	US9	34.70 ± 2.17 ^ab,B^	30.45 ± 4.20 ^b,A^	31.86 ± 2.27 ^a,AB^
Cn (g s)	NSU	360.25 ± 6.64 ^ab,A^	365.24 ± 4.46 ^c,A^	365.81 ± 3.49 ^a,A^
	US3	347.71 ± 16.48 ^a,B^	272.51 ± 23.98 ^ab,A^	339.85 ± 44.41 ^a,B^
	US6	384.48 ± 8.92 ^b,C^	254.59 ± 22.79 ^a,A^	321.80 ± 46.89 ^a,B^
	US9	356.80 ± 37.02 ^a,A^	309.66 ± 47.94 ^b,A^	322.20 ± 33.28 ^a,A^
Ch (g)	NSU	−5.07 ± 0.43 ^b,A^	−3.21 ± 0.30 ^d,B^	−5.03 ± 0.21 ^b,A^
	US3	−4.90 ± 0.39 ^b,B^	−7.27 ± 0.65 ^b,A^	−4.99 ± 0.66 ^b,B^
	US6	−6.29 ± 0.78 ^a,B^	−8.45 ± 0.75 ^a,A^	−6.10 ± 0.66 ^a,B^
	US9	−5.89 ± 0.72 ^a,A^	−5.04 ± 119 ^c,A^	−5.43 ± 0.76 ^ab,A^
*K* (Pa s*^n^*)	NSU	1.81 ± 0.56 ^a,B^	0.04 ± 0.03 ^a,A^	0.08 ± 0.01 ^a,A^
	US3	3.73 ± 1.11 ^a,B^	0.84 ± 0.46 ^b,A^	0.44 ± 0.04 ^a,A^
	US6	13.90 ± 3.51 ^b,B^	2.26 ± 0.52 ^b,A^	1.13 ± 0.16 ^b,A^
	US9	5.54 ± 0.80 ^a,B^	0.65 ± 0.18 ^ab,A^	0.48 ± 0.02 ^a,A^
*n*	NSU	0.48 ± 0.01 ^b,A^	0.44 ± 0.06 ^a,A^	0.75 ± 0.17 ^a,B^
	US3	0.43 ± 0.07 ^b,A^	0.67 ± 0.05 ^b,B^	0.62 ± 0.14 ^a,AB^
	US6	0.17 ± 0.03 ^a,A^	0.40 ± 0.08 ^a,B^	0.61 ± 0.03 ^a,C^
	US9	0.40 ± 0.03 ^b,A^	0.70 ± 0.01 ^b,B^	0.67 ± 0.09 ^a,B^

a–d Different lowercase letters indicate statistical differences between ultrasound treatments in goat milk yogurt samples during 28 days of storage (*p* < 0.05). A–C Different uppercase letters indicate statistical differences between days of storage in goat milk yogurt samples (*p* < 0.05). ^1^ F, firmness; Cn, consistency; Ch, cohesiveness; K, consistency index; n, flow behavior index. ^2^ NSU, non-sonicated goat milk yogurt; US3, goat milk yogurt sonicated for 3 min; US6, goat milk yogurt sonicated for 6 min; US9, goat milk yogurt sonicated for 9 min. Mean ± standard deviation from triplicate determinations.

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
