# Peer review of "Different Ultrasound Exposure Times Influence the Physicochemical and Microbial Quality Properties in Probiotic Goat Milk Yogurt"

_molecules, 2020, doi:10.3390/molecules25204638_

Round 1
Reviewer 1 Report
The manuscript reports on the application of ultrasound to modify the structure of goat milk yoghurt. It provides a couple of nice and interesting information. However, some revisions are necessary.
-L50: What are unusual consumers?
-L57-58: I do not understand this sentence. The “suitable ratio between additives and goat milk...” may be “variable, depending on the type of additive...”, but once the process is developed, it should not be a problem anymore.
-L67: The effect of ultrasound treatment on yoghurt texture is attributed to the denaturation of whey proteins, but this is also achieved with heat treatment (which additionally pasteurises the milk). What is the advantage of ultrasound compared to heat treatment?
-L69 and elsewhere: it should be “biogenic” rather than bioactive.
-L76-77: I am always a bit cautious about “health information” such as this because I find it very delicate to connect diseases and disorders such as schizophrenia, depression, and Parkinson to the consumption of yoghurt and the hypothetical presence of minor components that may or may not have been formed during processing. In fact, when looking at the given reference (#21; Benkerroum 2016), the reader just finds a table that compiles symptoms that have been observed for different biogenic amines, but it is difficult and time-consuming to identify the corresponding clinical studies from the reference list. In the main reference that is given by this author (Medina et al., 2003, Crit Rev Biochem Mol Biol 38:23), the reader does not find any connection between tyramine and schizophrenia, not to mention the absence of any information regarding (goat) yoghurt. It is unquestionable that biogenic amines can have a number of unpleasant effects on the human organism, but information like these given in a paper like this should be much more evidence-based regarding the actual hazardous potential of biogenic amines in yoghurt rather than citing a single paper that is much more focused on the individual compounds.
-L92: Please use food safety rather than food assurance.
-L157, L330: This is confusing because the ultrasound treatment was carried out after the actual fermentation.
-L416-438: How do you explain the contradictory effects of ultrasound and storage on firmness and viscosity?
-L416, L430: Costa et al. are publications from your own group. Are there also other studies to compare your results to?
-L420, L437: This is very speculative as there is no evidence in this research that ultrasound affects the “volume fraction of gel fragments” and “increases the concentration of casein micelles”. Furthermore, in the cited reference (#54, Wu et al. 2000), there is also no data for that. These authors just claimed this.
-L464: The activity of proteolytic enzymes should also affect the firmness, not only the viscosity.
-L468: “thickening properties” should lead to an increased viscosity.
-L485: What do 67% refer to?
-L492-493: On which basis were the ultrasound treatment times chosen?
-L512, L516, L518: Please use g/kg or similar rather than %
-L524, L529: What does “extracted” mean? Please give some more information on the sample preparation for these analyses.
-Section 4.7: Was any sample preparation done? Did you break and stir the yoghurt gels prior to the viscosity measurements?
-Figure 3: Please use the same scaling for the y-axes.
Author Response
GENERAL COMMENTS BY THE AUTHORS:
We performed revision following the previous comments done by reviewers. We believe that we have fully addressed all of these concerns and comments, which has increased the overall impact of the manuscript.
Changes are marked in yellow on the manuscript.
Reviewer #1:
The manuscript reports on the application of ultrasound to modify the structure of goat milk yoghurt. It provides a couple of nice and interesting information. However, some revisions are necessary..
AU: We appreciate the comment. We performed the proposed modifications.
Extensive editing of English language and style required.
AU: The manuscript was revised by native English writer. An revised version was resubmitted.
-L50: What are unusual consumers?
AU: Unusual or non-habitual consumers are those who do not consume determined food product frequently. Some manuscripts using the term “unusual consumers” or “non-habitual consumers” are reported below. However, the word “unusual” was deleted of the manuscript as required by reviewer 3. Page 2, line 49.
Costa, M. P., Balthazar, C. F., Franco, R. M., Mársico, E. T., Cruz, A. G., and Conte-Junior, C. A. (2014). Changes on expected taste perception of probiotic and conventional yogurts made from goat milk after rapidly repeated exposure. Journal of Dairy Science, 97(5): 2610-2618.
Costa, M. P., Frasao, B. S., Rodrigues, B. L., Silva, A. C. O., and Conte-Junior, C. A. (2016). Effect of different fat replacers on the physicochemical and instrumental analysis of low-fat cupuassu goat milk yogurts. Journal of Dairy Research, 83: 493-496.
Costa, M. P., Monteiro, M. L. G., Frasao, B. S., Silva, V. L. M., Rodrigues, B. L., Chiappini, C. C. J., and Conte-Junior, C. A. (2017). Consumer perception, health information, and instrumental parameters of cupuassu (Theobroma grandiflorum) goat milk yogurts. Journal of Dairy Science, 100: 1-12.
-L57-58: I do not understand this sentence. The “suitable ratio between additives and goat milk...” may be “variable, depending on the type of additive...”, but once the process is developed, it should not be a problem anymore.
AU: Thank you for your consideration. The purpose of this sentence in the introduction was to highlight that a suitable ratio between additives and goat's milk has not yet been achieved and that this limitation can be attributed to relationship among them being very variable. Thus, this is one of the reasons that reports in the literature have endeavored the development of innovative methods to improve the textural and rheological properties of goat´s milk yogurt. The sentence has been rewritten to make it clearer: Page 2, lines 52-54.
-L67: The effect of ultrasound treatment on yoghurt texture is attributed to the denaturation of whey proteins, but this is also achieved with heat treatment (which additionally pasteurises the milk). What is the advantage of ultrasound compared to heat treatment?
AU: When the thermosonication is performed in milk before inoculum with a starter culture, the resultant yogurt presents more viscous coagulum compared to yogurt elaborated from control milk (pasteurized milk) (Gursoy et al., 2016). This indicates that ultrasound is more efficient than pasteurization in promoting denaturation and consequent cross-linking between milk proteins, resulting in a firmer yogurt. In addition, the yogurt coagulum is formed during acidification of milk, when its pH reaches the isoelectric point of casein (pH 4.6) through fermentation with a starter culture, as described in the official method (AOAC, 2012). Thus, when ultrasound is employed after inoculum with a starter culture, the denaturation of whey proteins by ultrasound increases the associations of them with casein and casein micelles, consequently making the stronger yogurt coagulum. Pasteurization, however, has the limitation of cannot be used during fermentation or storage of yogurt. It because pasteurization will lead to the death of starter and probiotic cultures. Moreover, the high temperature in yogurt can change functional compounds and their sensory properties, which makes inadequate pasteurization during the manufacture and storage of yogurt. Consistently, in present work, the thermosonication was used with control of temperature at 8 ± 2 °C, to minimize the extent of lipid oxidation in yogurt (Page 15, lines 493-494).
References
AOAC International. Official Methods of Analysis, 19th ed.; AOAC Int.: Virginia, United States, 2012.
Gursoy, O.; Yilmaz, Y.; Gokce, O.; and Ertan, K. (2016). Effect of ultrasound power on physicochemical and rheological properties of yoghurt drink produced with thermosonicated milk. Emirates Journal of Food and Agriculture, 28(4): 235-241.
-L69 and elsewhere: it should be “biogenic” rather than bioactive.
AU: The word "bioctive" was changed to "biogenic" as suggested. Page 1, line 32; Page 2, line 66; Page 5, line 143; Page 7, lines 183, 184, and 210; Page 13, line 174; Page 15, line 470; Page 16, lines 548 and 549; Table 3.
-L76-77: I am always a bit cautious about “health information” such as this because I find it very delicate to connect diseases and disorders such as schizophrenia, depression, and Parkinson to the consumption of yoghurt and the hypothetical presence of minor components that may or may not have been formed during processing. In fact, when looking at the given reference (#21; Benkerroum 2016), the reader just finds a table that compiles symptoms that have been observed for different biogenic amines, but it is difficult and time-consuming to identify the corresponding clinical studies from the reference list. In the main reference that is given by this author (Medina et al., 2003, Crit Rev Biochem Mol Biol 38:23), the reader does not find any connection between tyramine and schizophrenia, not to mention the absence of any information regarding (goat) yoghurt. It is unquestionable that biogenic amines can have a number of unpleasant effects on the human organism, but information like these given in a paper like this should be much more evidence-based regarding the actual hazardous potential of biogenic amines in yoghurt rather than citing a single paper that is much more focused on the individual compounds.
AU: Thank you for your consideration. The neurological diseases were removed from the manuscript as suggested. Page 2, lines 71-72.
-L92: Please use food safety rather than food assurance.
AU: The term "food assurance" has been replaced by "food safety" as suggested. Page 2, line 91.
-L157, L330: This is confusing because the ultrasound treatment was carried out after the actual fermentation.
AU: Thanks for the comment. The sentences were rewritten to place the metabolization of monosaccharides and disaccharides in the context of storage. Page 6, lines 151-152; Page 12, lines 324-326.
-L416-438: How do you explain the contradictory effects of ultrasound and storage on firmness and viscosity?
AU: Ultrasound softens the protein gel because sound waves cause cavitation and strong shear forces in the fluid. Körzendörfer et al. (2019) reported that the maximum torque required to break the protein gel of Greek yogurt was reduced by 75% following the ultrasound, and gel firmness was reduced by 80%. In addition, sonication significantly increased the particle number, including fragments of gel, by a factor of approximately 2.5 in yogurts when compared to non-sonicated yogurts (Nöbel et al., 2016; Körzendörfer et al., 2017). Thus, sonication can result in weaker gels than respective control yogurts (Körzendörfer et al., 2017; Körzendörfer et al., 2019). In line, in the present work, in general, the firmness tended to reduce with ultrasound. However, this trend was not significant (p > 0.05) in most the cases (Table 4). On the other hand, yogurt gels produced from milk treated by high-intensity US have shown enhanced firmness (Nguyen and Anema, 2010; Nguyen et al., 2009; Nöbel et al., 2016; Riener et al., 2009; Riener et al., 2010; Sfakianakis and Tzia, 2014). This suggests that ultrasound application in milk instead of yogurt may be a better alternative to obtain a firmer product. On the other hand, as the particles number increases (Körzendörfer et al., 2017; Nöbel et al., 2016), ultrasound after inoculation improved viscosity in the finished yogurt (Wu et al. 2000), as also observed herein (Figure 3). This is explained in the manuscript (Pages 13-14, lines 410-421; Page 14, lines 436-438). Regard to the storage time only, the firmness (Table 4) and viscosity (Figure 3) reduced for all treatments of ultrasound as well as for control compared to fresh yogurt. This can be attributed to the release of proteolytic enzymes by starter cultures in yogurts during storage, which results in hydrolyzes of protein network to small peptides and free amino acids during storage, leading to significant reductions in texture (Gandhi et al., 2014). It is explained in the manuscript (Page 14, lines 423-425).
References
Gandhi, A., and Shah, N.P. (2014). Cell growth and proteolytic activity of Lactobacillus acidophilus, Lactobacillus helveticus, Lactobacillus delbrueckii ssp. bulgaricus, and Streptococcus thermophilus in milk as affected by supplementation with peptide fractions. International Journal of Food Sciences and Nutrition, 65(8): 937–941. Körzendörfer, A., Schäfer, J., Hinrichs, J., and Nöbel, S. (2019). Power ultrasound as a tool to improve the processability of protein-enriched fermented milk gels for Greek yogurt manufacture. Journal of Dairy Science, 102(9): 7826-7837.
Körzendörfer, A., Nöbel, S., and Hinrichs, J. (2017). Particle formation induced by sonication during yogurt fermentation – Impact of exopolysaccharide-producing starter cultures on physical properties. Food Research International, 97: 170-177.
Wu, H., Hulbert, G. J., and Mount, J. R. (2000). Effects of ultrasound on milk homogenization and fermentation with yogurt starter. Innovative Food Science & Emerging Technologies, 1(3): 211-218.
Nguyen, N.H.A.; Anema, S.G. Effect of ultrasonication on the properties of skim milk used in the formation of acid gels. Innov. Food Sci. Emerg. Technol. 2010, 11(4), 616-622. DOI: 10.1016/j.ifset.2010.05.006
Nguyen, T.M.P.; Lee, Y.K.; Zhou, W. Stimulating fermentative activities of bifidobacteria in milk by high intensity ultrasound. Int. Dairy J. 2009, 19(6-7), 410-416. DOI: 10.1016/j.idairyj.2009.02.004
Nöbel, S.; Ross, N.-L.; Protte, K.; Körzendörfer, A.; Hitzmann, B.; Hinrichs, J. Microgel particle formation in yogurt as influenced by sonication during fermentation. J. Food Eng. 2016, 180, 29-38. DOI: 10.1016/j.jfoodeng.2016.01.033
Riener, J.; Noci, F.; Cronin, D.A.; Morgan, D.J.; Lyng, J.G. The effect of thermosonication of milk on selected physicochemical and microstructural properties of yoghurt gels during fermentation. Food Chem. 2009, 114(3), 905-911. DOI: 10.1016/j.foodchem.2008.10.037
Riener, J.; Noci, F.; Cronin, D.A.; Morgan, D.J.; Lyng, J.G. A comparison of selected quality characteristics of yoghurts prepared from thermosonicated and conventionally heated milks. Food Chem. 2010, 119(3), 1108-1113. DOI: 10.1016/j.foodchem.2009.08.025
Sfakianakis, P.; Tzia, C. Conventional and Innovative Processing of Milk for Yogurt Manufacture; Development of Texture and Flavor: A Review. Foods 2014, 3(1), 176-193. DOI: 10.3390/foods3010176
L416, L430: Costa et al. are publications from your own group. Are there also other studies to compare your results to?
AU: Other studies were added to the discussion for comparison: Park et al., 2019, Miocinovic et al., 2016, Gursoy et al., 2016 and Li and Guo, 2006. Page 13, lines 407-409; Page 14, lines 427-431.
References
Park, Y. W., Oglesby, J., Hayek, S. A., Aljaloud, S. O., Gyawali, R., and Ibrahim, S. A. (2019). Impact of different gums on textural and microbial properties of goat milk yogurts during refrigerated storage. Foods, 8(169): 1-7.
Miocinovic, J., Miloradovic, Z., Josipovic, M., Nedeljkovic, A., Radovanovic, M., and Pudja, P. (2016). Rheological and textural properties of goat and cow milk set type yoghurts. International Dairy Journal, 58: 43-45.
Gursoy, O.; Yilmaz, Y.; Gokce, O.; and Ertan, K. (2016). Effect of ultrasound power on physicochemical and rheological properties of yoghurt drink produced with thermosonicated milk. Emirates Journal of Food and Agriculture, 28(4): 235-241.
Li, J., and Guo, M. (2006). Effects of polymerized whey proteins on consistency and water-holding properties of goat’s milk yogurt. Food Chemistry and Toxicology, 71(1): C34-C38.
L420, L437: This is very speculative as there is no evidence in this research that ultrasound affects the “volume fraction of gel fragments” and “increases the concentration of casein micelles”. Furthermore, in the cited reference (#54, Wu et al. 2000), there is also no data for that. These authors just claimed this.
AU: Thanks for the comment. The sentences were rewritten considering authors who effectively dosed gel particles number, firmness and viscosity in yogurts submitted to ultrasound compared to non-ultrasonicated yogurt (Pages 13-14, lines 410-414; Page 14, lines 437-439). In addition, the new references were added to the manuscript.
References
Körzendörfer, A., Nöbel, S., and Hinrichs, J. (2017). Particle formation induced by sonication during yogurt fermentation – Impact of exopolysaccharide-producing starter cultures on physical properties. Food Research International, 97: 170-177.
Körzendörfer, A., Schäfer, J., Hinrichs, J., and Nöbel, S. (2019). Power ultrasound as a tool to improve the processability of protein-enriched fermented milk gels for Greek yogurt manufacture. Journal of Dairy Science, 102(9): 7826-7837.
Nöbel, S., Ross, N.-L., Protte, K., Körzendörfer, A., Hitzmann, B., and Hinrichs, J. (2016). Microgel particle formation in yogurt as influenced by sonication during fermentation. J. Food Eng., 180: 29-38.
L464: The activity of proteolytic enzymes should also affect the firmness, not only the viscosity.
AU: The proteolytic activity of the starter culture over storage reduced both viscosity (Page 11, lines 267-269; Pages 14-15, lines 461-465) and firmness (Page 8, lines 229-232; Page 14, lines 421-426) for all ultrasound treatments as well as for control when compared with freshly prepared yogurt (Table 4).
L468: “thickening properties” should lead to an increased viscosity.
AU: Thanks for the comment. The sentence was corrected: “The increased flow behavior index in samples compared to the fresh yogurt (Table 4) may be related to the decrease of consistency index (K) values during storage, since both rheological parameters present inverse relation in pseudoplastic fluids (Krokida et al., 2007)”, Page 15, lines 465-468.
Reference
Krokida, M.K.; Maroulis, Z.B.; Saravacos, G.D. Rheological properties of fluid fruit and vegetable puree products: compilation of literature data. Int. J. Food Prop. 2007, 4(2), 179-200.
L485: What do 67% refer to?
AU: 67% refers to the amplitude of the ultrasound waves used in the experiment: “After fermentation, yogurt samples (150 mL) were subjected to US waves of 67% of amplitude in an ultrasonic bath at a frequency of 20 kHz”, Page 15, lines 489-490.
L492-493: On which basis were the ultrasound treatment times chosen?
AU: The ultrasound treatment times were chosen based on those reported in the literature for cow milk yogurt sonicated before (Riener et al., 2009; Riener et al., 2010) or after inoculum of starter cultures (Wu et al., 2001; Nöbel et al., 2016; Nguyen et al., 2009; Gholamhosseinpour and Hashem, 2019; Nguyen et al., 2012), as well as for unfermented cow and goat milk (Nguyen et al., 2010; Karlović, 2015). These information were added to the manuscript (Page 15, lines 498-502). In addition, ultrasound times over 15 minutes were reported to negatively affect textural properties of acid gels from cow milk (Nguyen et al., 2010).
References:
Riener, J.; Noci, F.; Cronin, D.A.; Morgan, D.J.; Lyng, J.G. The effect of thermosonication of milk on selected physicochemical and microstructural properties of yoghurt gels during fermentation. Food Chem. 2009, 114(3), 905-911.
Riener, J.; Noci, F.; Cronin, D.A.; Morgan, D.J.; Lyng, J.G. A comparison of selected quality characteristics of yoghurts prepared from thermosonicated and conventionally heated milks. Food Chem. 2010, 119(3), 1108-1113.
Wu, H.; Hulbert, G.J.; Mount, J.R. Effects of ultrasound on milk homogenization and fermentation with yogurt starter. Innov. Food Sci. Emerg. Technol. 2000, 1(3), 211–218.
Nöbel, S.; Ross, N.-L.; Protte, K.; Körzendörfer, A.; Hitzmann, B.; Hinrichs, J. Microgel particle formation in yogurt as influenced by sonication during fermentation. J. Food Eng. 2016, 180, 29-38.
Nguyen, T.M.P.; Lee, Y.K.; Zhou, W. Stimulating fermentative activities of bifidobacteria in milk by high intensity ultrasound. Int. Dairy J. 2009, 19(6-7), 410-416.
Gholamhosseinpour, A.; Hashemi, S.M.B. Ultrasound pretreatment of fermented milk containing probiotic Lactobacillus plantarum AF1: Carbohydrate metabolism and antioxidant activity. J. Food Process. Eng. 2019, 42(1), e12930.
Nguyen, T.M.P.; Lee, Y.K.; Zhou, W. Effect of high intensity ultrasound on carbohydrate metabolism of bifidobacteria in milk fermentation. Food Chem. 2012, 130(4), 866-874.
Nguyen, N.H.A.; Anema, S.G. Effect of ultrasonication on the properties of skim milk used in the formation of acid gels. Innov. Food Sci. Emerg. Technol. 2010, 11(4), 616-622.
Karlović, S. Reducing fat globules particle-size in goat milk: Ultrasound and high hydrostatic pressures approach. Chem. Biochem. Eng. Q. 2015, 28(4), 499-507.
L512, L516, L518: Please use g/kg or similar rather than %
AU: The "%" has been replaced by "g/Kg" in the lines mentioned by the reviewer. Page 3, line 111; Page 15, line 479; Page 16, lines 521, 524, 526 and 529.
L524, L529: What does “extracted” mean? Please give some more information on the sample preparation for these analyses.
AU: Biogenic amines, monosaccharides, disaccharides, and organic acids were extracted from yogurt samples. Description of the sample preparation and methods of analysis were added to the manuscript, as suggested by the reviewer. Page 16, lines 533-548; Pages 16-17, lines 553-569.
Section 4.7: Was any sample preparation done? Did you break and stir the yoghurt gels prior to the viscosity measurements?
AU: The yogurt gels were broken by stirring before of rheological measurements. This information was added to the manuscript. Page 17, lines 585-586.
Figure 3: Please use the same scaling for the y-axes.
AU: Thanks for the comment. The same scaling for the y-axes present in the graphics of figure 3 was performed, as suggested by reviewer.

Reviewer 2 Report
Additional remarks:
lines 86-93.: What is your hypothesis? Please add your hypothesis!
line 98. … first day of storage.... What does it mean? How many hours later was the microbial measuring after yogurt preparation?
line 118. I suggest that this sentence please put in the title.
Table 2.: the lactose values in the yogurt are relatively high (48-50 mg/g), in the references this value is appr. 30-40 mg/g. Please clarify!
uppercase letters: why used uppercase letters in the Table (such as LAC, US3) when no significance difference among treatment (storage time)? Please repair it!
Fig 1.: Total BA. What does it mean? Please add to the footnote! Bacterial names: please italicize!
line 164: What does LAB mean? Please add full words!
lines 208-209.: Bacterial names: please italicize!
line 385. this sentence not understand (600-50-6mg?), please correct it!
line 477. 4.2 subsection: How much time done when reached the final pH? Do you use cooling (final temperature?) to prevent the over-acidification? or yogurt samples subjected to US waves at 43 0C immediately? How much time done when applied the US treatment? But in Fig 2: NSU pH change was start at appr. pH 4.3! Please clarify!
line 479.: Bacterial name: please italicize!
line 508. 4.5.1 subsection: why not use chemical composition (or approximate) instead of proximate composition
line 519. pH is not chemical trait! This is a physical attribution!
Author Response
GENERAL COMMENTS BY THE AUTHORS:
We performed revision following the previous comments done by reviewers. We believe that we have fully addressed all of these concerns and comments, which has increased the overall impact of the manuscript.
Changes are marked in yellow on the manuscript.
Reviewer #2
English language and style are fine/minor spell check required
AU: The article was sent for English editing by a native English writer, and the revised version was resubmitted.
lines 86-93.: What is your hypothesis? Please add your hypothesis!
AU: Thanks for the comment. The hypothesis was added to the manuscript. Page 2, lines 89-91.
line 98. … first day of storage.... What does it mean? How many hours later was the microbial measuring after yogurt preparation?
AU: The first day of storage means 24h after the yogurt preparation. Therefore, the microbial measuring was performed in the refrigerated yogurt (at 4 °C) 24h after the yogurt preparation. This information was added to the manuscript: Page 3, lines 96-97.
line 118. I suggest that this sentence please put in the title.
AU: The sentence in the line 118 (Page 3) is as follows: “Mean ± standard deviation from triplicate determinations.” We believe that there may have been a typo by the reviewer when mentioned the line number of the sentence that should be added to the title. We kindly ask for the reviewer to ratify or rectify the line number whose sentence should be added to the title.
Table 2.: the lactose values in the yogurt are relatively high (48-50 mg/g), in the references this value is appr. 30-40 mg/g. Please clarify!
AU: Thank you for the comment. Mba, Boyo, and Oyenuga (1975) showed that the lactose content in goat milk used for yogurt manufacture varied from 44.3 to 59.7 mg/g, depending on the breed and stages of lactation of dairy goat. In line, Bagnicka, Łukaszewicz, and Ådnøy (2016) demonstrated that breed, parity and litter size affecting the lactose content, which varied from 45.5 to 46.2 mg/g in goat milk. In an Austrian study, the lactose content varied from 40.6 to 48.8 mg/g also in goat milk (Mayer and Fiechter, 2012). In the reference cited in our work (Costa et al. 2016), the content of lactose in goat milk was of 58.92 mg/g by HPLC (High Performance Liquid Chromatography). After 4h of fermentation, the lactose content varied from 40.16 to 44.99 mg/g in fermented goat´s milks (Costa et al. 2016). Coherently, the lactose contents by other studies were reported as being 45.3 mg/g (Khaled, Illek, and Gajdò·Ek, 1999), 41 mg/g (Rojo et al., 2015), 46.6 mg/g (Marín et al., 2011), and 41.6 mg/g (Kapadiya et al., 2016) in the goat milk. Therefore, the lactose content in unfermented goat milk by several studies varied from 40.6 to 59.7 mg/g. Hence, the lactose content in the yogurts in our work (48-50 mg/g) is within the range described in the literature for goat´s milk. However, the lactose values (40.16 to 44.99 mg/g) in fermented goat milks in work referenced by us were slightly lower than those reported by us (48-50 mg/g) in goat milk yogurts. This can be justified by possible differences between both studies regarding the lactose content in the unfermented milk used in the manufacture of fermented dairy products. This justification was added to the manuscript: Page 12, lines 310-317.
References:
Bagnicka, E., Łukaszewicz, M., and Ådnøy, T. (2016). Genetic parameters of somatic cell score and lactose content in goat’s milk. Journal of Animal and Feed Sciences, 25: 210-215. DOI: 10.22358/jafs/65552/2016.
Costa, M. P., Frasao, B. S., Lima, B. R. C. C., Rodrigues, B. L., and Conte-Júnior, C. A. (2016). Simultaneous analysis of carbohydrates and organic acids by HPLC-DAD-RI for monitoring goat's milk yogurts fermentation. Talanta, 152: 162-170. DOI: 10.1016/j.talanta.2016.01.061.
Kapadiya, D.B., Prajapati, D. B., Jain, A. K., Mehta, B. M., Darji, V. B., and Apharnati, K. D. (2016). Comparison of Surti goat milk with cow and buffalo milk for gross composition, nitrogen distribution, and selected minerals content. Veterinary World, 9(7): 710-716. DOI: 10.14202/vetworld.2016.710-716.
Khaled, N. F., Illek, J., and Gajdò·Ek, S. (1999). Interactions between nutrition, blood metabolic profile and milk composition in dairy goats. Acta Veterinaria Brno, 68: 253-258. DOI: 10.2754/avb199968040253.
Mayer, H. K., and Fiechter, G. (2012). Physical and chemical characteristics of sheep and goat milk in Austria. International Dairy Journal, 24(2): 57-63. DOI: 10.1016/j.idairyj.2011.10.012
Mba, A. U., Boyo, B. S., and Oyenuga, V. A. (1975). Studies on the milk composition of West African dwarf, Red Sokoto and Saanen goats at different stages of lactation. Journal of Dairy Research, 42: 217-226. DOI: 10.1017/s0022029900015259.
Rojo, R., Kholif, A. E., Salem, A. Z. M., Elghandour, M. M. Y., Odongo, N. E., Oca, R. M., Rivero, N., and Alonso, M. U. (2015). Influence of cellulase addition to dairy goat diets on digestion and fermentation, milk production and fatty acid content. Journal of Agricultural Science, 153: 1514-1523. DOI: 10.1017/S0021859615000775.
Table 2. uppercase letters: why used uppercase letters in the Table (such as LAC, US3) when no significance difference among treatment (storage time)? Please repair it!
AU: Thank you for the comment. The correction was performed, as suggested. Uppercase and lowercase letters were maintained only for significant differences (p < 0.05) among storage days and among treatments, respectively. Table 2.
Fig 1.: Total BA. What does it mean? Please add to the footnote! Bacterial names: please italicize!
AU: Thank you for the comment. The meaning of "total BA" (total biogenic amines) was added to the footnote of Figure 1 (Page 5, line 143). The bacterial names in the footnote of Figure 1 were italicized as required (Page 5, lines 140-141).
line 164: What does LAB mean? Please add full words!
AU: Thank you for the comment. The meaning of "LAB" (lactic acid bacteria) was added to the manuscript. Page 6, line 158.
lines 208-209.: Bacterial names: please italicize!
AU: Thank you for the comment. The bacterial names were italicized as required. Page 7, line 200.
line 385. this sentence not understand (600-50-6mg?), please correct it!
AU: Thank you for the comment. The sentence was corrected to improve its understanding. Page 13, lines 376-381.
line 477. 4.2 subsection: How much time done when reached the final pH? Do you use cooling (final temperature?) to prevent the over-acidification? or yogurt samples subjected to US waves at 43 0C immediately? How much time done when applied the US treatment? But in Fig 2: NSU pH change was start at appr. pH 4.3! Please clarify!
AU: Thank you for the comment. The final pH was reached after 4 h of fermentation. This information was added to the manuscript – Page 15, lines 485-486. After 4 h of fermentation, the final pH (4.4) was reached. Then, fermentation was stopped by refrigerating the yogurts at 4 ± 1°C. After 2 h of refrigeration, the yogurt samples at 4 ± 2 °C were subjected to the ultrasound treatments – Page 15, lines 486-488. The figure 2 shows the pH changes in the yogurt samples during refrigerated storage at 4 ± 1°C, in the intervals of 1, 14 and 28 days of storage. The 1st day of storage means 24 h after the yogurt manufacture - Page 3, lines 96-97. Therefore, although the final pH in the yogurt samples immediately after fermentation has been 4.4, post-acidification occurs during storage, since β-galactosidase remains active even at refrigerated storage temperature (0-5 °C) (Kailasapathy et al., 2008) – Page 12, lines 333-336. As in the non-sonicated samples (NSU) the viability of lactic acid bacteria was greater compared to sonicated samples (Table 1), the post-acidification in NSU was more dramatic, leading to a pH reduction of 4.4 to 4.29 after 1 day of storage.
Reference:
Kailasapathy, K.; Harmstorf, I.; Phillips, M. Survival of Lactobacillus acidophilus and Bifidobacterium animalis ssp. lactis in stirred fruit yogurts. LWT-Food Sci. Technol. 2008, 41(7), 1317-1322. DOI: doi.org/10.1016/j.lwt.2007.08.009.
line 479.: Bacterial name: please italicize!
AU: Thank you for the comment. The bacterial name was italicized as required. Page 15, line 482.
line 508. 4.5.1 subsection: why not use chemical composition (or approximate) instead of proximate composition
AU: The “proximate composition” term was replaced by chemical composition in 4.5.1 section (Page 16, line 520), as well as over manuscript (Page 3, lines 107, 111, 115, 116, and 118; Page 11, lines 304-305, and 307; Page 15, lines 479; Page 16, lines 516 and 520; Page 18, line 619).
line 519. pH is not chemical trait! This is a physical attribution!
AU: Thank you for the comment. The subsection “4.6.1. pH determination” was inserted in section “4.6. Physical characterization of goat milk yogurt”. Page 17, line 576.

Reviewer 3 Report
Comments and recommendations - for example:
- 50 – to change or delete the word unusual
- 77-84 – the part should be improved
- 103, 290 – species
- 110, 114, 120 etc. – what does mean proximate? It would be more appropriate “chemical composition”
- 124, 125, 133 etc. – why carbohydrates? It would be more appropriate “mono- and disaccharides”
- Tables 1 and 2 are too complicated….in Table 1 and 2: it has not to be divided to BC and PC or CHO and OA, it is written in a caption of Table 1 and 2
- 1 – the X-axis should be the same for all groups (for example 0 – 12)
- 145, 146, 208, 209, 479 etc. - the names of microorganisms should be written in Italic
- P or p – please, unify lower and uppercase letters and these abbreviations should be written in Italic
- 2 – for this type of results should be better using of column, not line
- Please correct the units for Cn, K (for example Table 4)
- Please, explain the low count of probiotic microorganisms in control samples
- 428, 468 – Str. thermophiles, not thermophiles
- 477-483 it would be appropriate to indicate the composition of the raw milk
- 493-495 it does not belong to this chapter (US treatments)
- please, simplify the chapters – for example l. 509 …Chemical composition was done according official methods……..
- References: the same citation 19 and 20 (Linares et al. 2012); 13 and 30 (Nguyen et al. 2009);
- I think that language and style of English must to be improved
Author Response
GENERAL COMMENTS BY THE AUTHORS:
We performed revision following the previous comments done by reviewers. We believe that we have fully addressed all of these concerns and comments, which has increased the overall impact of the manuscript.
Changes are marked in yellow on the manuscript.
Reviewer #3:
50 – to change or delete the word unusual
AU: Thank you for the comment. The word “unusual” was deleted. Page 2, line 49.
77-84 – the part should be improved
AU: Thank you for the comment. The part was improved for better understanding of information. Page 2, lines 71-81.
103, 290 – species
AU: The word was corrected as required. Page 3, line 101; Page 11, line 281.
110, 114, 120 etc. – what does mean proximate? It would be more appropriate “chemical composition”
AU: The “proximate composition” term was replaced by chemical composition in 4.5.1 section (Page 16, line 520), as well as over manuscript (Page 3, lines 107, 111, 115, 116, and 118; Page 11, lines 304-305, and 307; Page 15, lines 479; Page 16, lines 516 and 520; Page 18, line 619).
124, 125, 133 etc. – why carbohydrates? It would be more appropriate “mono- and disaccharides”
AU: Thank you for the comment. The term “carbohydrates” was replaced by “mono- and disaccharides as required. Page 3, lines 120 and 121; Page 4, lines 129 and 134; Page 15, line 473; Page 16, lines 530, 531, 534, and 540.
Tables 1 and 2 are too complicated….in Table 1 and 2: it has not to be divided to BC and PC or CHO and OA, it is written in a caption of Table 1 and 2
AU: Thank you for the comment. The Tables 1 and 2 have been simplified as suggested.
1 – the X-axis should be the same for all groups (for example 0 – 12)
AU: Thank you for the comment. The X-axis was standardized (from 0 to 12 microbial counts) for all groups as suggested. Figure 1.
145, 146, 208, 209, 479 etc. - the names of microorganisms should be written in Italic
AU: Thank you for your consideration. The bacterial names were written in Italic as required. Page 1, line 28; Page 3, lines 100, 101, and 102; Page 5, lines 140-141; Page 7, line 200; Page 11, line 276 and 297; Page 13, lines 362, 364 and 366; Page 14, lines 424-425; Page 15, lines 482, 486, and 508; Page 16, line 511.
P or p – please, unify lower and uppercase letters and these abbreviations should be written in Italic
AU: Thank you for your consideration. The probability symbol was standardized in lower letter and Italic (p). Page 3, line 110 and 118; Page 4, lines 132 and 133; Page 5, line 139; Page 6, line 162; 165, 166, 173, and 180; Page 7, lines 186, 192, 193, 198, 200, 204, and 208; Page 8, lines 213, 214, 229, and 232; Page 9, lines 236 and 237; Page 11, line 266; Page 14, line 416; Page 18, lines 620 and 622.
2 – for this type of results should be better using of column, not line
AU: The format of graph has been changed from row to column as suggested. Figure 2.
Please correct the units for Cn, K (for example Table 4)
AU: Units for consistency and consistency index have been corrected as required. Table 4. Page 8, lines 222, 223, 224, and 225; Page 10, lines 258, 259, and 260; Page 17, line 596.
Please, explain the low count of probiotic microorganisms in control samples
AU: The counts of starter cultures (Streptococcus thermophilus and Lactobacillus bulgaricus) in our work were 10.30 and 7.44 log CFU g-1 (Table 1) in the control yogurt samples. Therefore, the counts of starter cultures for control samples herein were higher than 7 Log CFU g-1, what characterizes the product as yogurt (Codex Alimentarius, 2010). Moreover, the count of the probiotic L. acidophilus LA-5 was 8.05 Log CFU g-1 (Table 1) in the control yogurt. Therefore, the count of L. acidophilus LA-5 was greater than the minimum count needed for the characterization of the product as a probiotic, which is 6 Log CFU g-1 (Shah et al., 2000).
References:
- Codex Alimentarius. 2010. Codex standard for fermented milks. 2nd ed. Codex standard 243–2003 in Codex Alimentarius: Milk and Milk Products. Codex Alimentarius Commission, Brussels, Belgium.
- Shah, N. P., Ali, J. F., & Ravula, R. R. (2000). Populations of Lactobacillus acidophilus, Bifidobacterium spp., and Lactobacillus casei in commercial fermented milk products. Bioscience Microflora, 19(1), 35-39. DOI: 10.12938/bifidus1996.19.35.
428, 468 – Str. thermophiles, not thermophiles
AU: The correction was performed as required. Page 14, line 425.
477-483 it would be appropriate to indicate the composition of the raw milk
AU: Thanks for the comment. The chemical composition and pH value of the goat´s milk used in the manufacture of yogurt samples were added to the manuscript. Page 15, lines 478-480.
493-495 it does not belong to this chapter (US treatments)
AU: Thanks for the comment. The part was distributed in the appropriate sections. Page 16, lines 513-514 and 516-519; Page 17, lines 573-575.
please, simplify the chapters – for example l. 509 …Chemical composition was done according official methods
AU: The sessions were not simplified because the other reviewers requested that the methodology be more detailed. Thus, more details were added to simplified sessions at the request of the other reviewers. Page 15, lines 487-480, 485-488, 499-503; Page 16, lines 534-549; Pages 16-17, lines 554-570; Page 17, line 586.
References: the same citation 19 and 20 (Linares et al. 2012); 13 and 30 (Nguyen et al. 2009)
AU: Thanks for the comment. The citations were corrected as required.
I think that language and style of English must to be improved
AU: The manuscript was revised by native English writer. A revised version was resubmitted.

Round 2
Reviewer 1 Report
L71-81: The health aspects are still too generic. Please provide specific information for (goat) yoghurt or remove this part
Author Response
GENERAL COMMENTS BY THE AUTHORS:
We performed a new revision of the manuscript following the recent comments done by reviewers. We believe that we have fully addressed all of these concerns and comments, which has increased the overall impact of the manuscript.
Changes are marked in yellow on the manuscript.
Reviewer #1:
L71-81: The health aspects are still too generic. Please provide specific information for (goat) yoghurt or remove this part
AU: We appreciate the comment. We added references considering the effects of ingestion of each amine (tyramine and polyamines) specifically through the consumption of yogurt on health aspects of human volunteers. Pages 2-3, lines 71-108.
References:
- Baratella, D.; Bonaiuto, E.; Magro, M.; Roger, J. A.; Kanamori, Y.; Lima, G. P. P.; Agostinelli, E.; Vianello, F. Endogenous and food‑derived polyamines: determination by electrochemical sensing. Amino acids 2018, 50(9), 1187-1203. DOI: 10.1007/s00726-018-2617-4
- Benamouzig, R.; Mahé, S.; Luengo, C.; Rautureau, J.; Tomé, D. Fast and postprandial polyamine concentrations in the human digestive lumen. J. Clin. Nutr. 1997, 65(3), 766-770. DOI: 10.1093/ajcn/65.3.766
- Büyükuslu, N. Dietary polyamines and diseases: reducing polyamine intake can be beneficial in cancer treatment. Nutr. 2015, 2(2), 27-38. DOI: 10.18488/journal.87/2015.2.2/87.2.27.38
- Fan, J.; Feng, Z.; Chen, N. Spermidine as a target for cancer therapy. Res. 2020, 159, 104943. DOI: 10.1016/j.phrs.2020.104943
- Hsu, C. C.; Chow, W-H.; Boffetta, P.; Moore, L.; Zaridze, D.; Moukeria, A.; Janout, V.; Kollarova, H.; Bencko, V.; Navratilova, M.; Szeszenia-Dabrowska, N.; Mates, D.; Brennan, P. Dietary risk factors for kidney cancer in eastern and central Europe. J. Epidemiol. 2007, 166(1), 62-70. DOI: doi.org/10.1093/aje/kwm043
- Kesse, E.; Bertrais, S.; Astorg, P.; Jaouen, A.; Arnaultu, N.; Galan, P.; Hercberg, S. Dairy products, calcium and phosphorus intake, and the risk of prostate cancer: results of the French prospective SU.VI.MAX (Supple´mentation en Vitamines et Mine´raux Antioxydants) study. J. Nutr. 2006, 95, 539-545. DOI: 10.1079/BJN20051670
- Kumar, S.; Kumar, B.; Kumar, R.; Kumar, S.; Khatkar, S. K.; Kanawjia, S. K. Nutritional features of goat milk – A review. Indian J. Dairy Sci. 2012, 65(4), 266-273. DOI: 10.5146/IJDS.V65I4.25762.G11905
- López-Plaza, B.; Bermejo, L. M.; Santurino, C.; Cavero-Redondo, I.; Álvarez-Bueno, C.; Gómez-Candela, C. Milk and dairy product consumption and prostate cancer risk and mortality: an overview of systematic reviews and meta-analyses. Nutr. 2019, 10(2), S212-S223. DOI: 10.1093/advances/nmz014
- Matsumoto, M.; Benno, Y. Consumption of Bifidobacterium lactis LKM512 yogurt reduces gut mutagenicity by increasing gut polyamine contents in healthy adult subjects. Res. 2004, 568, 147-153. DOI: 10.1016/j.mrfmmm.2004.07.016
- Matsumoto, M.; Ohishi, H.; Benno, Y. Impact of LKM512 yogurt on improvement of intestinal environment of the elderly. FEMS Immunol. Med. Microbiol. 2001, 31(3), 181-186. DOI: doi.org/10.1111/j.1574-695X.2001.tb00518.x
- McCabe-Sellers, B. J.; Staggs, C. G.; Bogle, M. L. Tyramine in foods and monoamine oxidase inhibitor drugs: A crossroad where medicine, nutrition, pharmacy, and food industry converge. Food Compos. Anal. 2006, 19, S58-S65. DOI: 10.1016/j.jfca.2005.12.008
- Saikali, J.; Picard, C.; Freitas, M.; Holt, P. Fermented milks, probiotic cultures, and colon cancer. Cancer 2004, 49(1), 14-24. DOI: 10.1207/s15327914nc4901_3
- Tadjine, D. Milk heat treatment affects microbial characteristics of cows’ and goats’ “Jben” traditional fresh cheeses. Food Sci. Technol. 2020, 1-8. DOI: 10.1590/fst.00620
- Zheng, X.; Wu, K.; Song, M.; Ogino, S.; Fuchs, C. S.; Chan, A. T.; Giovannucci, E. L.; Cao, Y.; Zhang, X. Yogurt consumption and risk of conventional and serrated precursors of colorectal cancer. Gut 2020, 69(5), 970-972. DOI: 10.1136/gutjnl-2019-318374

Reviewer 3 Report
l. 140 - correct Bulgaricus
l. 427 - correct thermophiles
Author Response
Reviewer #3
- 140 - correct Bulgaricus
AU: Thank you for the comment. Bulgaricus was changed for bulgaricus – Page 6, line 167.
- 427 - correct thermophiles
AU: Thank you for the comment. thermophiles was changed for thermophilus – Page 15, line 455.
